# Deep learning enhanced Rydberg multifrequency microwave recognition

Zong-Kai Liu [1,2], Li-Hua Zhang[1,2], Bang Liu[1,2], Zheng-Yuan Zhang[1,2], Guang-Can Guo[1,2], Dong-Sheng Ding [1,2 ✉] & Bao-Sen Shi [1,2 ✉]

Recognition of multifrequency microwave (MW) electric fields is challenging because of the complex interference of multifrequency fields in practical applications. Rydberg atom-based measurements for multifrequency MW electric fields is promising in MW radar and MW communications. However, Rydberg atoms are sensitive not only to the MW signal but also to noise from atomic collisions and the environment, meaning that solution of the governing Lindblad master equation of light-atom interactions is complicated by the inclusion of noise and high-order terms. Here, we solve these problems by combining Rydberg atoms with deep learning model, demonstrating that this model uses the sensitivity of the Rydberg atoms while also reducing the impact of noise without solving the master equation. As a proof-of-principle demonstration, the deep learning enhanced Rydberg receiver allows direct decoding of the frequency-division multiplexed signal. This type of sensing technology is expected to benefit Rydberg-based MW fields sensing and communication.

[1] Key Laboratory of Quantum Information, University of Science and Technology of China, Hefei, Anhui 230026, China. [2] Synergetic Innovation Center of Quantum Information and Quantum Physics, University of Science and Technology of China, Hefei, Anhui 230026, China. ✉email: dds@ustc.edu.cn; drshi@ustc.edu.cn

The strong interaction between Rydberg atoms and microwave (MW) fields that results from their high polarizability means that the Rydberg atom is a candidate medium for MW fields measurement, e.g., using electromagnetically induced absorption[1], electromagnetically induced transparency (EIT)[2,3] and the Autler–Townes effect[3–6]. The amplitudes[7–10], phases[10,11] and frequencies[9,10] of MW fields could then be measured with high sensitivity. Based on this measurement sensitivity for MW fields, the Rydberg atom has been used in communications[7,8,12,13] and radar[14] as an atom-based radio receiver. In the communications field, the Rydberg atom replaces the traditional antenna with superior performance aspects that include sub-wavelength size, high sensitivity, system international (SI) traceability to Planck's constant, high dynamic range, self-calibration and an operating range that spans from MHz to THz frequencies[7,9,10,15,16]. One application is analogue communications, e.g., real-time recording and reconstruction of audio signals[13]. Another application is digital communications, e.g., phase-shift keying and quadrature amplitude modulation[7,8,12]. The channel capacity of MW-based communications is limited by the standard quantum limited phase uncertainty[7]. Furthermore, a continuously tunable radio-frequency carrier has been realized based on Rydberg atoms[17], thus paving the way for concurrent multichannel communications. Detection and decoding of multifrequency MW fields are highly important in communications for acceleration of information transmission and improved bandwidth efficiency. Additionally, MW fields recognition enables simultaneous detection of multiple targets with different velocities from the multifrequency spectrum induced by the Doppler effect. However, because of the sensitivity of Rydberg atoms, the noise is superimposed on the message, meaning that the message cannot be recovered efficiently. Additionally, it is difficult to generalize and scale the band-pass filters to enable demultiplexing of multifrequency signals with more carriers[16].

To solve these problems, we use a deep learning model for its accurate signal prediction capability and its outstanding ability to recognize complex information from noisy data without use of complex circuits. The deep learning model updates the weights via backpropagation and then extracts features from massive data without human intervention or prior knowledge of physics and the experimental system. Because of these advantages, physicists have constructed complex neural networks to complete numerous tasks, including far-field subwavelength acoustic imaging[18], value estimation of a stochastic magnetic field[19], vortex light recognition[20,21], demultiplexing of an orbital angular momentum beam[22,23] and automatic control of experiments[24–29].

Here, we demonstrate a deep learning enhanced Rydberg receiver for frequency-division multiplexed digital communication. In our experiment, the Rydberg atoms act as a sensitive antenna and a mixer to receive multifrequency MW signals and extract information[9,11,12]. The modulated signal frequency is reduced from several gigahertz to several kilohertz via the interaction between the Rydberg atoms and the MWs, thus allowing the information to be extracted using simple apparatus. These interference signals are then fed into a well-trained deep learning model to retrieve the messages. The deep learning model extracts the multifrequency MW signal phases, even without knowing anything about the Lindblad master equation, which describes the interactions between atoms and light beams in an open system theoretically. The solution of the master equation is often complex because the higher-order terms and the noises from the environment and from among the atoms are taken into consideration. However, the deep learning model is robust to the noise because of its generalization ability, which takes advantage of the sensitivity of the Rydberg atoms while also reducing the impact of the noise that results from this sensitivity. Our deep

learning model is scalable, allowing it to recognize the information carried by more than 20 MW bins. Additionally, when the training is complete, the deep learning model extracts the phases more rapidly than via direct solution of the master equation.

## Results

**Setup.** We adapt a two-photon Rydberg-EIT scheme to excite atoms from a ground state to a Rydberg state. A probe field drives the atomic transition $|5S_{1/2}, F = 2\rangle \rightarrow |5P_{1/2}, F' = 3\rangle$ and a coupling light couples the transition $|5P_{1/2}, F' = 3\rangle \rightarrow |51D_{3/2}\rangle$ in rubidium 85, as shown in Fig. 1a. Multifrequency MW fields drive a radio-frequency (RF) transition between the two different Rydberg states $|51D_{3/2}\rangle$ and $|50F_{5/2}\rangle$. The energy difference between these states is 17.62 GHz. The multifrequency MW fields consist of multiple MW bins (more than three bins) with frequency differences of several kilohertz from the resonance frequency. The amplitudes, frequencies, and phases of the multiple MW bins can be adjusted individually (further details are provided in the "Methods" section). The detunings of the probe, coupling and MW fields are $\Delta_p$, $\Delta_c$ and $\Delta_s$, respectively. The Rabi frequencies of the probe, coupling and MW fields are $\Omega_p$, $\Omega_c$ and $\Omega_s$, respectively. The experimental setup is depicted in Fig. 1b. We use MW fields to drive the Rydberg states constantly, producing modulated EIT spectra, i.e., the probe transmission spectra, as shown in the inset of Fig. 1b. The phases of the MW fields correlate with the modulated EIT spectra and can be recovered from these spectra with the aid of deep learning. Specifically, the probe transmission spectra are fed into a well-trained deep learning model that consists of a one-dimensional convolution layer (1D CNN), a bi-directional long–short-term memory layer (Bi-LSTM) and a dense layer to extract the phases of the MW fields. Figure 1c–e shows these components of the neural network (further details are presented in the "Methods" section). Finally, the bin phases are recovered and the data are read out.

**Frequency-division multiplexed signal encoding and receiving.** In the experiments, we use a four-bin frequency-division multiplexing (FDM) MW signal for demonstration, where one of the four MW bins is used as the reference bin. The relative phase differences between the reference bin and the other bins are modulated by the message signal. Specifically, for the four-bin MW signal,

$$E = A_1 \cos[(\omega_0 + \omega_1)t + \varphi_1] + A_2 \cos[(\omega_0 + \omega_2)t + \varphi_2] \\ + A_3 \cos[(\omega_0 + \omega_3)t + \varphi_3] + A_4 \cos[(\omega_0 + \omega_4)t + \varphi_4],$$

where $\omega_0$ is the resonant frequency, $\omega_{1,2,3}$ are the relative frequencies, the carrier frequencies are $2\pi(\omega_0 + \omega_1) = 17.62\,\text{GHz} - 3\,\text{kHz}$, $2\pi(\omega_0 + \omega_2) = 17.62\,\text{GHz} - 1\,\text{kHz}$, $2\pi(\omega_0 + \omega_3) = 17.62\,\text{GHz} + 1\,\text{kHz}$ and $2\pi(\omega_0 + \omega_4) = 17.62\,\text{GHz} + 3\,\text{kHz}$, the frequency difference between two frequency-adjacent bins is $\Delta f = 2\,\text{kHz}$ and the message signal is $\varphi_{1,2,3} = 0$ or $\pi$, standing for 3 bits (0 or 1), and the reference phase is $\varphi_4 = 0$ (which remains unchanged). The phase list $(\varphi_1, \varphi_2, \varphi_3, \varphi_4)$ is a bit string for time $t_0$. By varying the phase of $\varphi_{1,2,3}$ with time, we then obtain the FDM signal for binary phase-shift keying (2PSK). Additionally, the amplitudes of the four bins are $0.1A_4 = A_{1,2,3}$ to solve the problem that results from the nonlinearity of the atom, where the probe transmission spectra of two different bit strings, e.g. $(0, 0, \pi, 0)$ and $(0, \pi, 0, 0)$, are the same (further details are presented in the "Methods" section). By increasing the frequency difference $\Delta f$, we can obtain higher information transmission rates. For four bins with $\Delta f = 2\,\text{kHz}$, the information transmission rate is $n_b \times \Delta f = (4 - 1) \times 2 \times 10^3\,\text{bps} = 6\,\text{kbps}$, where $n_b$ is the number of bits. In the experiments, disturbances originate from

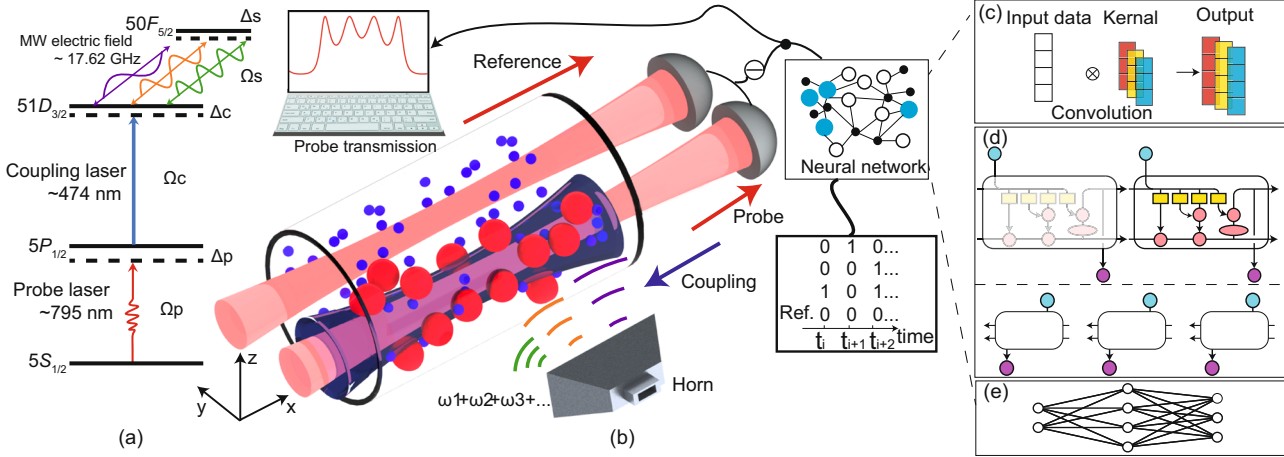

**Fig. 1 Illustration of the setup. a** Overview of experimental energy diagram. Probe and coupling laser beams excite the atoms at ground state $|5S_{1/2}\rangle$ to the Rydberg state $|51D_{3/2}\rangle$. Multifrequency microwave (MW) electric fields couple the Rydberg states $|51D_{3/2}\rangle$ and $|50F_{5/2}\rangle$. **b** Schematic of Rydberg atom-based antenna and mixer interacting with multifrequency signals. A 795 nm laser beam is split into two beams, which then propagate in parallel through a heated Rb cell (length: 10 cm, temperature: 44.6 °C, atomic density: $9.0 \times 10^{10}$ cm$^{-3}$)[46]. One is the probe beam, which counterpropagates with the coupling laser beam exciting atoms to Rydberg states to reduce Doppler broadening. The other is the reference beam, which does not counterpropagate with the coupling laser beam. The beams are detected using a differencing photodetector (DD) to obtain the probe transmission spectrum (inset). Multifrequency MW fields transmitted by a horn are applied to the atoms, with a radiated direction that is perpendicular to the laser beam propagation direction. The multifrequency MW fields are modulated using a phase signal such that the phase differences between the reference bin and the other bins carry the messages. The probe transmission spectrum is fed into a well-trained neural network to retrieve the variations of the phases with time. **c–e** Schematics of the neural network. The network consists of **c** a one-dimensional convolution layer, **d** a bi-directional long–short-term memory layer and **e** a dense layer; for further details about these layers, see the "Methods" section.

the environment and atomic collisions. Because of the sensitivity of Rydberg atoms to MW fields, the resulting noise submerges our signal. To use the sensitivity of the Rydberg atoms and simultaneously minimize the effects of noise, the deep learning model is used to extract the relative phases $(\varphi_1, \varphi_2, \varphi_3)$.

**Deep learning**. To improve the robustness and speed of our receiver, we use a deep learning model to decode the probe transmission signal. The complete encoding and decoding process is illustrated in Fig. 2a. The Rydberg antenna receives the FDM-2PSK signal and down-converts this signal into the probe transmission spectrum. The information is then retrieved from the spectrum using the deep learning model. The precondition is that the different bit strings correspond to distinct probe spectra; this is resolved by setting $0.1A_4 = A_{1,2,3}$, as discussed earlier. Then, we combine the 1D CNN layer, the Bi-LSTM layer and the dense layer to form the deep learning model (see the "Methods" section for further details)[30,31]. One of the reasons for using the 1D CNN layer and the Bi-LSTM layer is that the data sequences are long, which means that prediction of the phases $\varphi = (\varphi_1, \varphi_2, \varphi_3, 0)$ from the spectrum is a regression task and requires a long-term memory for our model. Another reason is to combine the convolution layer's speed with the sequential sensitivity of the Bi-LSTM layer[32]. The input sequence is first processed by the 1D CNN to extract the features, meaning that a long sequence is converted into a shorter sequence with higher-order features. This process is visualized to show how the deep learning model treats the transmission spectrum; more details are presented in the Supplementary Materials. The shorter sequence is then fed into the Bi-LSTM layer and resized by the dense layer to match the label size (see the "Methods" section for further details). Specifically, the probe spectrum $\mathbf{T} = \{T_0, T_\tau, T_{2\tau}, \cdots, T_{i\cdot\tau}\cdots, T_{N\cdot\tau}\}$ and the corresponding phases $\boldsymbol{\varphi} = (\varphi_1, \varphi_2, \varphi_3, \varphi_4 = 0)$ are collected to form the data set, where $T_{i\cdot\tau}$ is the $i$th data point of a probe spectrum and the fourth

bit $\varphi_4 = 0$ is the reference bit. Both the spectra and the phases are 1D vectors with dimensions of $N+1$ and 4, respectively. These independent, identically distributed data $\{\{\mathbf{T}\}, \{\boldsymbol{\varphi}\}\}$ are fed into our model as a data set. By shuffling this data set and splitting it into three sets, i.e., a training set, a validation set and a test set, we train our model on the training set (feeding both the waveforms and labels $\{\{\mathbf{T}\}, \{\boldsymbol{\varphi}\}\}$), validate, and test our model on the validation and test sets, respectively (by feeding waveforms without labels and comparing the predictions with ground truth labels). The validation set is used to determine whether there is either overfitting or underfitting during training. Finally, the performance (i.e., accuracy) of the model is estimated by predicting the test set.

The performance of our deep learning model is affected by the training epochs and the training and validation set sizes. The training curves on different training sets and validation sets are shown in Fig. 2b, c. Initially, our model performs well on the training set only, implying overfitting. The curves then converge (dashed line) and our model performs well on both the training set and the validation set. The sudden jump in the loss curve in Fig. 2c is caused by the change in the learning rate (further details are presented in the "Methods" section). Use of more training and validation data causes the curves to converge more quickly. The deep learning model performs well after these few-sample training. In Fig. 2d, we show a confusion matrix for prediction of a uniformly distributed test set, which demonstrates accuracy of 99.38%.

The "noise" shown in Fig. 2a refers to two kinds of noises. One comes from atoms and the external environment (systematic noise). The other comes from the noise added on purpose (additional noise). The systematic noise cannot be adjusted quantitatively and is discussed with its noise spectrum in the Supplementary Materials. Because the noise on the data set is independent and is distributed identically (i.i.d.), i.e., the entire data set is shuffled before being split into the training and test sets, the systematic noise pattern is almost the same in both the training set and the test set. The deep learning model has already

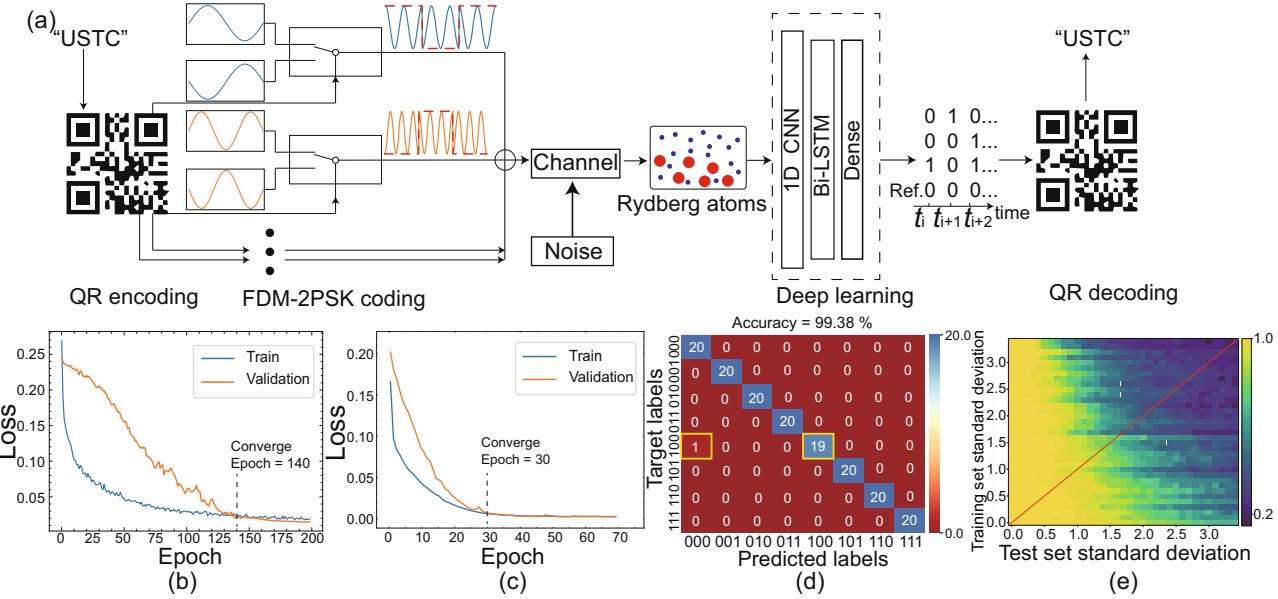

**Fig. 2 Flow chat of recognizing the multifrequency MWs and results. a** Process of encoding and decoding frequency-division multiplexed binary phase-shift keying (FDM-2PSK) signal with Rydberg atoms and the deep learning model. **b, c** Loss curve evolution with epochs for the training set (blue) and the validation set (orange) during training with different training data and validation data. The training data sizes are **b** 393 and **c** 1194. The validation data sizes are **b** 131 and **c** 398; more details about the data set split are presented in the "Methods" section. The loss curves for training and validation converge at **b** 140 and **c** 30 epochs, where an epoch is a time unit during which the model iterates once over the complete data set; see "Methods" section. **d** Confusion matrix for a test set (the number of the testing set is 160 and the labels are uniformly distributed) after training in case (**c**). The accuracy reaches 99.38% after a 70-epoch training period. **e** Deep learning model accuracy on the noisy test set after training on the noisy training set. The x- and y-axes represent the standard deviations of the additional white noise added to the test set and the training set, respectively. The colorbar represents the accuracy of the model on the noisy test set. The results were obtained by averaging five sets of predictions. The diagonal (red line) indicates the accuracy of the model on a test set in which the noise distribution is the same as that of the training set; more details about the noise are shown in Supplementary Materials.

learned the systematic noise pattern during the training process, which is one of the major advantages of use of deep learning against systematic noise. However, there is a case where the noise is not i.i.d. (i.e., the case where a specific noise occurs during testing only). This problem can be solved by online learning and addition of prior knowledge as new features into the data, e.g., data for the temperature, the weather, and other factors[33]. Here for simplicity, we talk about the i.i.d. case only and add the white noise with a mean $\mu$ and a standard deviation $\sigma$. We ignore the $1/f$ noise in this case because it decays rapidly in the low frequency range and the signal with which it would interfere is located within the 2–200 kHz range. The additional noise is added both on the training set and the test set of the deep learning model in Fig. 2e, which demonstrates the performance of the deep learning model when used on a data set with biased or unbalanced noise. The results below the red line show the performance of the model after training on a weaker-noise training set when predicting based on a stronger-noise test set, i.e., generalization for a stronger noise case. These results indicate that the deep learning model has the generalization ability required to adapt to stronger noise. In the area above the red line, there is more noise in the training set than in the test set. Theoretically, a small amount of additional noise in the training set will increase the robustness of the deep learning model. However, when the noise increases, it affects the accuracy, which decays rapidly. Next, the well-trained model is used to reconstruct the QR code. In Fig. 3a–c, the results and the corresponding confusion matrices with their epochs are shown. First, the information is encoded into a QR code. After the code is transmitted, received and decoded using the Rydberg atoms and the deep learning model, the information is then reconstructed successfully using the 35-epoch training model in Fig. 3c, but is not reconstructed in parts (a) and (b). The accuracy is defined by the number of correctly predicted bit strings divided

by the total number of bit strings (147 bit strings). After 35 epochs, the accuracy reaches 99.32% and the message is reconstructed successfully from the QR code received.

**Comparison between deep learning method and the master equation.** In our case, the master equation that we employed is the commonly used one without considering the noise spectrum. The accuracies of the deep learning model and the master equation fitting on noisy data are different. Figure 4 shows the accuracies obtained by the two methods. The deep learning model is trained on a training set without additional noise, and tested on a test set with additional white noise whose standard deviation is $\sigma$ (the transmission spectra with noise are given in Supplementary Materials). Here for simplicity, the data set is composed of the transmission of four MW bins only (one of them is reference bin) and the frequency difference between the adjoin bins is $\Delta f = 2$kHz. On the other hand, the result of the master equation is given based on the same test set as that of the deep learning model. The deep learning method outperforms the fitting of the master equation on the noisy data set.

Apart from the robustness to the noise, when the transmission rate is increased by increasing the number of MW bins or the frequency difference $\Delta f$, the deep learning model performs well, while it is difficult to retrieve the messages with high accuracy using the master equation. Specifically, to increase the bandwidth efficiency and the transmission rate, the number of MW bins used to carry the messages must be increased, but the information is still recognizable because of the scalability of the deep learning model. For 20 MW bins, the number of bits is $(20-1)$ with one reference bit, giving a $(20-1) \times 2$ kbps $= 38$ kbps transmission rate. The number of combinations of these bits is $2^{19}$, which increases exponentially as the number of MW bins increases.

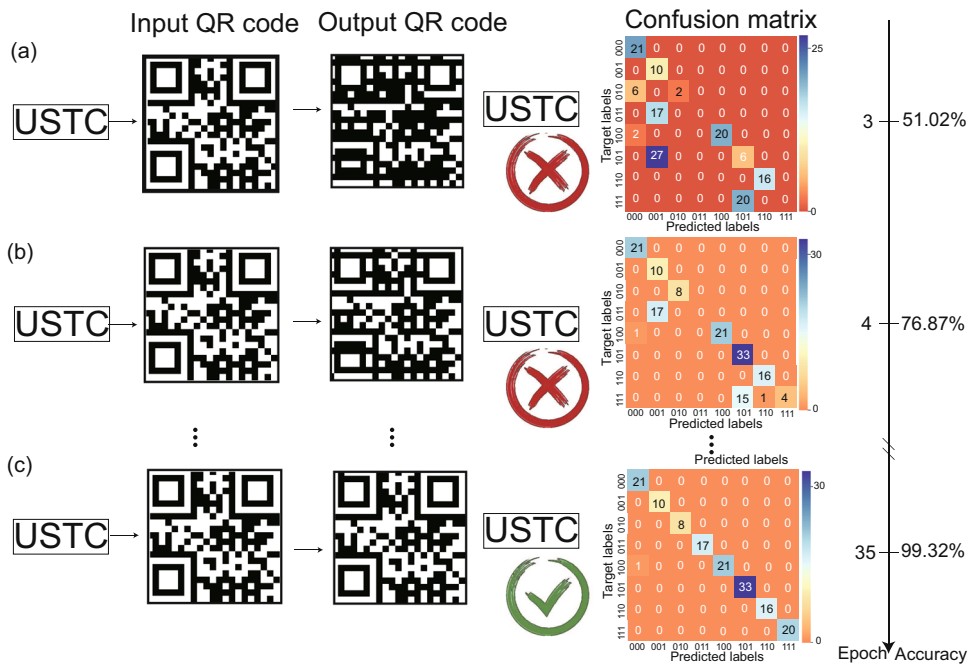

**Fig. 3 Reconstruction of the QR code after different epochs of training of the deep learning model.** After **a** 3 epochs, **b** 4 epochs, and **c** 35 epochs of training, the accuracies of our deep learning model are 51.02%, 76.87%, 99.32%, respectively, as determined from the confusion matrices (last column). The y-axis is the ground truth and the x-axis represents the predictions of the deep learning model. The colour bar and the numbers indicate the counts. If the prediction is correct, then the corresponding diagonal term in the confusion matrix increases by 1. If incorrect, then the nondiagonal element increases by 1.

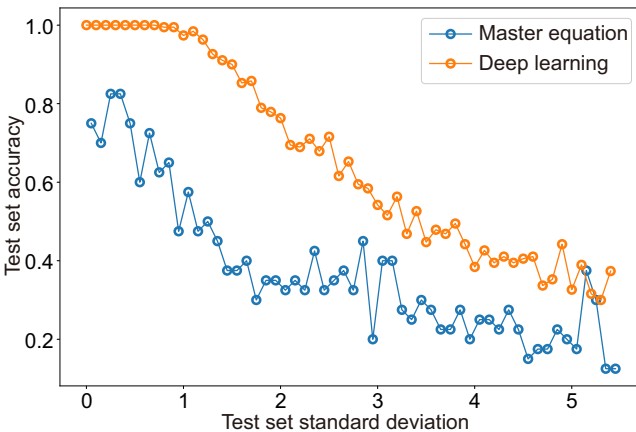

**Fig. 4 Prediction accuracy of deep learning model and the master equation on the noisy test set.** The noise is white noise with mean $\mu = 0$ and a standard deviation $\sigma$. The noise is added quantitatively using different $\sigma$ values. Before and after addition of the noise, the data are scaled between 0 and 1 using their maximum and minimum. The deep learning model is trained in a training set without additional noise. In addition, we do not involve noise spectrum when solving the master equation. Each point is obtained after averaging of five predictions on the noisy data.

Here, for demonstration purposes, only the first 3 bits of the total of 19 bits carry the messages and the other bits, including the reference, are set to be 0. To show how well our model performs, we train, validate and test the model on this new data set without varying the other parameters, with the exception of the training epochs of our model. The loss curves for training and validation are shown in Fig. 5a. A confusion matrix for epoch 78 is shown in Fig. 5b. The model performs well on this new test set, which was sampled uniformly from eight categories with an accuracy of 100%. Another method that can be used to increase the

information transmission rate involves increasing the frequency difference. In our case, the frequency difference is increased from $\Delta f = 2$ kHz to $\Delta f = 200$ kHz. The transmission rate increases correspondingly, from $(4 - 1) \times 2$ kbps = 6 kbps to $(4 - 1) \times 200$ kbps = 0.6 Mbps. To detect the high-speed signal, the DD bandwidth is increased, which inevitably leads to increased noise. After the model is trained on this new data set, the training and validation loss curves are as shown in Fig. 5c. A confusion matrix for epoch 83 is shown in Fig. 5d. Increasing the number of training epochs allows the model to perform well on this new data set, with an accuracy of 98.83% on a uniformly sampled test set.

To compare the performances of the deep learning model and the master equation, we fitted the probe spectra for 20 bins with a frequency difference $\Delta f = 2$ kHz and four bins with a frequency difference $\Delta f = 200$ kHz by solving the master equation without considering the higher-order terms and the effects of noise. In each case, 160 probe spectra were fitted that were sampled uniformly from every category. The prediction results are shown in Fig. 5(e) and (f). The prediction accuracy of the master equation is lower than that of the deep learning model. In our case, the impact of increasing the number of bins is greater than increasing the DD bandwidth for high-speed signals on the fitting accuracy. The prediction accuracy for a 20-bin carrier with frequency difference $\Delta f = 2$ kHz is 20.63%, which is like to the accuracy of guessing, i.e., 1/8. This implies that there is a disadvantage that comes from the fitting method itself, i.e., it can easily become trapped by local minima. Some type of prior knowledge is required to overcome this disadvantage, e.g., provision of the initial values of the phases before fitting. In contrast, the deep learning model is data driven and does not require any prior knowledge. The local minima problem of deep learning can be overcome using some well-known techniques, including learning rate scheduling and design of a more effective optimizer[32]. Additionally, the accuracy difference for the 200-

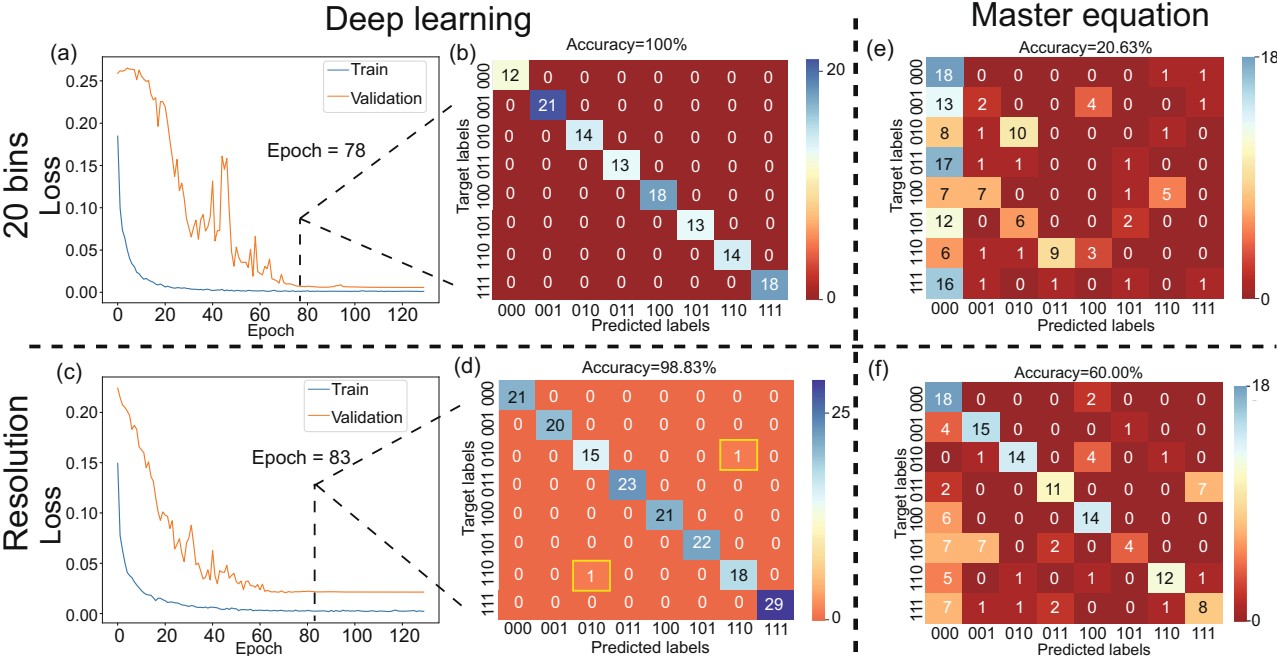

**Fig. 5 Loss curves and confusion maps of the deep learning method and the master equation fitting method for MWs with 20 bins and frequency difference Δf. a** Loss vs. epoch curves for training and validation of a 20-bin carrier with frequency difference $\Delta f = 2$ kHz. **b** Confusion matrix for epoch = 78 on the test data set. Only the first 3 bits of the 20 bits are selected to carry the messages and all other bits are 0. The accuracy is 100% for 123 testing spectra. **c** Loss vs. epoch curves for training and validation of the four MW bins (one bin is the reference) with frequency difference $\Delta f = 200$ kHz. **d** Confusion matrix for epoch = 83 on the test data set. The accuracy is 98.83% for 171 testing spectra. **e** Predicted solution of the master equation without consideration of higher-order terms and noise on a 20-bin carrier with frequency difference $\Delta f = 2$ kHz. The number of spectra is 160, where the spectra were sampled uniformly from eight categories. The accuracy is 20.63%. **f** Predicted solution of the master equation without consideration of higher-order terms and noise on four-bin carrier with frequency difference $\Delta f = 200$ kHz. The number of spectra is 160, which were sampled uniformly from eight categories. The accuracy is 60.00%. In (**b**) and (**e**), only the first 3 bits of a total of 20 bits (including a reference) are labelled.

kHz-difference MW bins between the deep learning model and the master equation means that the deep learning model is more robust to noise. Furthermore, the prediction time for the master equation is 25 s per spectrum, while the time for the deep learning model is 1.6 ms per spectrum. The master equation is solved by "FindFit" function in Mathematica 11.1 with both "Accuracy-Goal" and "PrecisionGoal" default, while the deep learning code is written in Python 3.7.6. These codes are run on the same computer with NVIDIA GTX 1650 and Intel®Core™ i7-9750H.

Another method to decode the signal is available that uses an in-phase and quadrature (I–Q) demodulator or a lock-in amplifier[7,12]. However, the carrier frequency must be given when decoding the signal in this case. Additionally, for multiple MW bins, numerous bandpass filters are required. The deep learning method is thus much more convenient.

## Discussion
We report a work on Rydberg receiver enhanced via deep learning to detect multifrequency MW fields. The results show that the deep learning enhanced Rydberg mixer receives and decodes multifrequency MW fields efficiently; these fields are often difficult to decode using theoretical methods. Using the deep learning model, the Rydberg receiver is robust to noise induced by the environment and atomic collisions and is immune to the distortion that results from the limited bandwidths of the Rydberg atoms (from dipole-dipole interactions and the EIT pumping rate, as studied in ref. [7]) for high-speed signals ($\Delta f = 200$ kHz). In addition to increasing the transmission speed of the signals, further increments in the information transmission rate are achieved by using more bins, which is feasible because of

the scalability of our model. Besides the transmission rate, this deep learning enhanced Rydberg system promises for use in studies of the channel capacity limitations. Because spectra that are difficult for humans to recognize as a result of noise and distortion are distinguishable when using the deep learning model, Rydberg systems enhanced by deep learning could take steps toward the realization of the capacity limit proposed in the literature ref. [34]. To obtain high performance (i.e. high signal-to-noise ratio, information transmission rate, channel capacity and accuracy), the training epochs and training set must be extended and enlarged.

In summary, we have demonstrated the advantages of receiving and decoding multifrequency signals using a deep learning enhanced Rydberg receiver. In a multifrequency signal receiver, rather than using multiple band-pass filters, lock-in amplifier[7,12] and other complex circuits, signals can be decoded using the extremely sensitive Rydberg atoms and the deep learning model at high speed and with high accuracy without solving the Lind-blad master equation. One of the advantages of use of the Ryd-berg atom is that the accuracy of the Rydberg atom approaches the photon shot noise limit[35]. In principle, the accuracy of the Rydberg atom is higher than that of the classical antenna. According to recent work based on the atomic superheterodyne method, ultrahigh sensitivity can be obtained[10]. However, in this proof-of-principle demonstration, there is considerable room for the optimization required to reach that limit (e.g., stabilization of the laser, narrowing the laser linewidth, and temperature stabi-lization). The sensitivity of the Rydberg atoms is a double-edged sword because it also involves noise. The deep learning model restricts this side effect while taking full advantage of the Rydberg atoms' sensitivity to the signal. Using the automatic feature

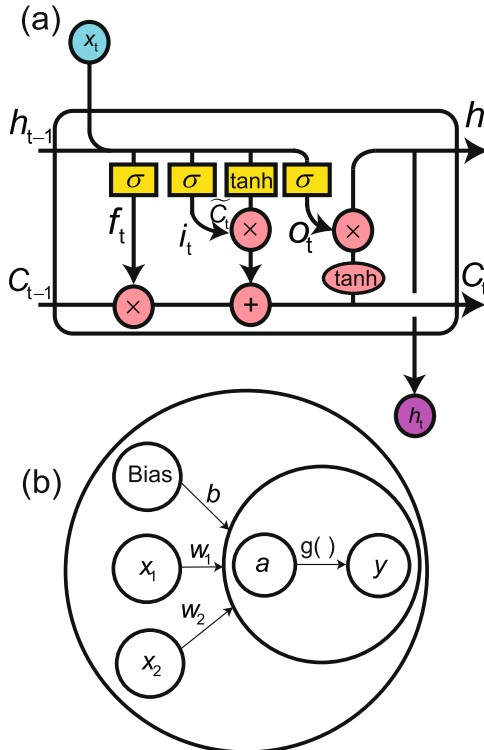

**Fig. 6 Building blocks for LSTM layer and dense layer.** The cells of a long–short-term memory (LSTM) layer and dense layer are presented in (**a**) and (**b**), respectively, where $\sigma$ represents the sigmoid layer and tanh represents the tanh layer.

extraction processes of the neural networks, the spectra are classified in a supervised manner. If the features (e.g. mean value, variance, frequency spectrum) are extracted manually, the spectra are then clustered by unsupervised learning methods such as t-distributed stochastic neighbour embedding (t-SNE) or the density-based spatial clustering of applications with noise (DBSCAN) method[31], without training on the training set. Our work will be useful in fields including high-precision signal measurement and atomic sensors. Additionally, this decoding ability can be generalized further to decode other signals that are encoded by different encoding protocols, e.g., frequency division multiplexing amplitude shift keying (FDM-ASK), frequency division multiplexing quadrature amplitude modulation (FDM-QAM), and IEEE 802.11ac WLAN standard signals for a 5 GHz carrier. The frequency of carrier to be decoded covers from several hertz to terahertz since for Rydberg atoms to receive MW with different wavelengths, the only part of the system that needs to be tuned is the frequency of the laser, while in classical receivers, the wavelength of the received MW is limited by the size of the antenna[36–39]. In addition to communications, our receiver can be used to detect multiple targets from multi-frequency signals caused by the Doppler effect.

## Methods

**Generation and calibration of MW fields**. The MW fields used in our experiments were synthesized by the signal generator (1465F-V from Ceyear) and a frequency horn. Each bin in the multifrequency MW field is tunable in terms of frequency, amplitude and phase. The RF source operates in the range from DC to 40 GHz. The frequency horn is located close to the Rb cell. We used an antenna and a spectrum analyser (4024F from Ceyear) to receive the MW fields and then calibrated the amplitudes of the MW fields at the centre of the Rb cell.

The probe transmission spectrum in the time domain when $\Delta_p = 0$, $\Delta_c = 0$ and $\Delta_s = 0$ reflects the interference among the multifrequency MW bins, which results from the beat frequencies of the bins that occur through the interaction between the atoms and light. The Rydberg atoms receive the MW bins by acting as an

antenna and a mixer[9,11,12]. After reception by the atoms, the frequency spectrum of the probe transmission shows that we can obtain the frequency differential signal from the probe transmission spectrum. This represents an application of our atoms to reduce the modulated signal frequency (from terahertz to kilohertz magnitude), which allows the signal to be received and decoded using simple apparatus. In our experiment, more than 20 frequency bins can be added to the atoms, for which the dynamic range is greater than 30 dBm. The amplitudes, phases and frequencies of these bins can be tuned individually. When the bandwidth is increased to detect an increasing frequency difference $\Delta f$ signal, more noise is involved, but this noise is suppressed by the deep learning model. In other words, the signal can be recognized using the deep learning model when the information transmission rate is increased by raising the frequency difference $\Delta f$. These bins are used to send FDM-PSK signals in the "FDM signal encoding and receiving" section of the main text.

**Master equation**. The Lindblad master equation is given as follows: $d\rho/dt = -i[H, \rho]/\hbar + L/\hbar$, where $\rho$ is the density matrix of the atomic ensemble and $H = \sum_k H[\rho^{(k)}]$ is the atom–light interaction Hamiltonian when summed over all the single-atom Hamiltonians using the rotating wave approximation. This Hamiltonian has the following matrix form:

$$H = \hbar \begin{pmatrix} 0 & -\frac{\Omega_p}{2} & 0 & 0 \\ -\frac{\Omega_p}{2} & \Delta_p & -\frac{\Omega_c}{2} & 0 \\ 0 & -\frac{\Omega_c}{2} & \Delta_c + \Delta_p & -\frac{\Omega_s(t)}{2} \\ 0 & 0 & -\frac{\Omega_s(t)}{2} & \Delta_c + \Delta_p + \Delta_s \end{pmatrix}, \tag{1}$$

where for the MW signal $E = A_1 \cos[(\omega_0 + \omega_1)t + \varphi_1] + A_2 \cos[(\omega_0 + \omega_2)t + \varphi_2] + A_3 \cos[(\omega_0 + \omega_3)t + \varphi_3] + A_4 \cos[(\omega_0 + \omega_4)t + \varphi_4]$, we have the Rabi frequency $\Omega_s(t) = \sqrt{E_1^2 + E_2^2}$, where $E_1 = A_1 \sin[\omega_1 t + \varphi_1] + A_2 \sin[\omega_2 t + \varphi_2] + A_3 \sin[\omega_3 t + \varphi_3] + A_4 \sin[\omega_4 t + \varphi_4]$ and $E_2 = A_1 \cos[\omega_1 t + \varphi_1] + A_2 \cos[\omega_2 t + \varphi_2] + A_3 \cos[\omega_3 t + \varphi_3] + A_4 \cos[\omega_4 t + \varphi_4]$. The Rabi frequency can be derived as follows:

$$\begin{aligned} E &= \sum_{i=1}^{4} A_i \cos\left[(\omega_0 + \omega_i)t + \varphi_i\right] \\ &= \sqrt{E_1^2 + E_2^2} \cos\left(\omega_0 t + \arctan\frac{\sum_{i=1}^{4} A_i \sin(\omega_i t + \varphi_i)}{\sum_{i=1}^{4} A_i \cos(\omega_i t + \varphi_i)}\right), \end{aligned} \tag{2}$$

where the second term (which resonates with the energy levels of the Rydberg atoms) induces the normal EIT spectrum and the first term modulates that spectrum. In the interaction between the atoms and the MW fields, the atoms act as a mixer such that the output signal frequency ($\omega_1$, $\omega_2$, $\omega_3$) is less than the input signal frequency ($\omega_0 + \omega_1$, $\omega_0 + \omega_2$, $\omega_0 + \omega_3$). The modulation signal's nonlinearity is reduced by setting the reference and increasing its amplitude as shown in Eq. (3), which is a precondition for recognition of these phases via deep learning.

$$\begin{aligned} &\sqrt{E_1^2 + E_2^2} \\ &\approx A_4 \sqrt{1 + 2\sum_{i=1}^{3} \frac{A_i}{A_4} \cos\left[(\omega_4 - \omega_i)t + (\varphi_4 - \varphi_i)\right]} \\ &\approx A_4 + \sum_{i=1}^{3} A_i \cos\left[(\omega_4 - \omega_i)t + (\varphi_4 - \varphi_i)\right], \end{aligned} \tag{3}$$

where the condition for the approximations on the second line and the third line is $A_4 \gg A_{1,2,3}$.

The Lindblad superoperator $L = \sum_k L[\rho^{(k)}]$ is composed of single-atom superoperators, where $L[\rho^{(k)}]$ represents the Lindbladian and has the following form: $\frac{L[\rho^{(k)}]}{\hbar} = -\frac{1}{2}\sum_m (C_m^\dagger C_m \rho + \rho C_m^\dagger C_m) + \sum_m C_m \rho C_m^\dagger$ where $C_1 = \sqrt{\Gamma_e}|g\rangle\langle e|$, $C_2 = \sqrt{\Gamma_r}|e\rangle\langle r|$ and $C_3 = \sqrt{\Gamma_s}|r\rangle\langle s|$ are collapse operators that stand for the decays from state $|e\rangle$ to state $|g\rangle$, from state $|r\rangle$ to state $|e\rangle$ and from state $|s\rangle$ to state $|r\rangle$ with rates $\Gamma_e$, $\Gamma_r$ and $\Gamma_s$, respectively. Because we are only concerned with the steady state here, i.e. $t \to \infty$, the Lindblad master equation can be solved using $d\rho/dt = 0$. The complex susceptibility of the EIT medium has the form $\chi(\nu) = (|\mu_{ge}|^2/\varepsilon_0\hbar)\rho_{eg}$, where $\rho_{eg}$ is the element of density matrix solved using the master equation. The spectrum of the EIT medium can be obtained from the susceptibility using $T \sim e^{-Im[\chi]}$.

**Deep learning layers**. Our deep learning model consists of a 1D CNN layer, a Bi-LSTM layer and a dense layer. The mathematical sketches for these layers are given as follows.

The 1D CNN layer is illustrated in Fig. 1c. The input signal convolutes the kernel in the following form:

$$f \otimes g = \sum_{m=0}^{N-1} f_m g_{(n-m)}. \tag{4}$$

where $f$ represents the input data, $g$ is the convolution kernel, $m$ is the input data index and $n$ is the kernel index. The 1D CNN extracts the higher-order features

from the input data to reduce the lengths of the sequences fed into the Bi-LSTM layer. Before flowing into the Bi-LSTM layer, the data pass through the batch normalization layer, the ReLU activation layer and the max-pooling layer, in that sequence. For a mini-batch $\mathcal{B} = \{x_{1\cdots m}\}$, the output from the batch normalization layer is $y_i = BN_{\gamma,\beta}(x_i)$ and the learning parameters are $\gamma$ and $\beta$[40]. The update rules for the batch normalization layer are:

$$\mu_{\mathcal{B}} \leftarrow \frac{1}{m}\sum_{i=1}^{m} x_i, \tag{5}$$

$$\sigma_{\mathcal{B}}^2 \leftarrow \frac{1}{m}\sum_{i=1}^{m} \left(x_i - \mu_{\mathcal{B}}\right)^2, \tag{6}$$

$$\hat{x}_i \leftarrow \frac{x_i - \mu_{\mathcal{B}}}{\sqrt{\sigma_{\mathcal{B}}^2 + \epsilon}}, \tag{7}$$

$$y_i \leftarrow \gamma\hat{x}_i + \beta \equiv BN_{\gamma,\beta}(x_i), \tag{8}$$

where Eqs. (5) and (6) evaluate the mean and the variance of the mini-batch, respectively; the data are normalized using the mean and the variance in Eq. (7) and the results are then scaled and shifted in Eq. (8). The training is accelerated using the batch normalization layer and the overfitting is also weakened by this layer. The output then passes through the ReLU activation layer. The activation function of this layer is $f_{\text{ReLU}}(x) = \max(x, 0)$. The vanishing gradient problem is diminished by this activation function. Next, the inputs are downsampled in a max-pooling layer[30].

The LSTM layer and an LSTM cell are shown schematically in Figs. 1d and 6a, respectively. The equations for the LSTM are shown as Eqs. (9)–(14)[32,41]. At a time $t$, the input $x_t$ and two internal states $C_{t-1}$ and $h_{t-1}$ are fed into the LSTM cell. The first thing to be decided by the LSTM cell is whether or not to forget in Eq. (9), which outputs a number between 0 and 1 that represents retaining or forgetting. Next, an input gate (Eq. (10)) decides which values are to be updated from a vector of new candidate values created using Eq. (11). The new value is then added to the cell state and the old value is forgotten in Eq. (12). Finally, the cell decides what to output using Eqs. (13) and (14).

$$f_t = \sigma\left(W_f \cdot [h_{t-1}, x_t] + b_f\right), \tag{9}$$

$$i_t = \sigma\left(W_i \cdot [h_{t-1}, x_t] + b_i\right), \tag{10}$$

$$\tilde{C}_t = \tanh\left(W_C \cdot [h_{t-1}, x_t] + b_C\right), \tag{11}$$

$$C_t = f_t \times C_{t-1} + i_t \times \tilde{C}_t, \tag{12}$$

$$o_t = \sigma\left(W_o[h_{t-1}, x_t] + b_o\right), \tag{13}$$

$$h_t = o_t \times \tanh\left(C_t\right), \tag{14}$$

where $\sigma(x) = 1/(1 + e^{-x})$ is the sigmoid function. The sigmoid and tanh functions are applied in an element-wise manner. The LSTM is followed by a time-reversed LSTM to constitute a Bi-LSTM layer that improves the memory for long sequences.

The dense layer and a neuron are drawn in Figs. 1e and 6b, respectively, and the corresponding equations are

$$\mathbf{a} = \mathbf{w} \cdot \mathbf{x} + b, \tag{15}$$

$$\mathbf{y} = g(\mathbf{a}), \tag{16}$$

where $\mathbf{w}$ is the vector of weights, $b$ is the bias, $x$ represents the input data, $g(\mathbf{a}) = 1/(1 + e^{-\mathbf{a}})$ is the sigmoid activation function used to limit the output values to between 0 and 1, and $y$ is the output. The dense layer resizes the shape of the data obtained from the Bi-LSTM to match the size of the label.

The training consists of both forward and backward propagation. A batch of probe spectra propagates through the 1D CNN layer, the Bi-LSTM layer, and dense layer during the forward training process. The differentiable loss function is then calculated. In our case, the differentiable loss function is the mean squared error (MSE) between the predictions and the ground truth, which is used widely in the regression task[32]. The equation for the MSE is

$$L_{\text{MSE}} = \frac{1}{m \cdot n}\sum_{i=1}^{n}\sum_{j=1}^{m}\left(\varphi_{i,j} - f(T_{i,j})\right)^2, \tag{17}$$

where $m$ is the number of data points in one spectrum, $n$ is the mini-batch size, $\varphi_i$ is the ground truth and $f(T_i)$ is the model prediction. In backpropagation, the trainable weights of each layer are updated based on the learning rate and the derivative of the MSE loss function with respect to the weights to minimize the loss $L_{\text{MSE}}$, such that

$$W \leftarrow W - \eta\frac{\partial L_{\text{MSE}}}{\partial W}, \tag{18}$$

where $\eta$ is the learning rate and $W$ is the trainable weight for each layer. The weights of each layer are then updated according to the RMSprop optimizer[42].

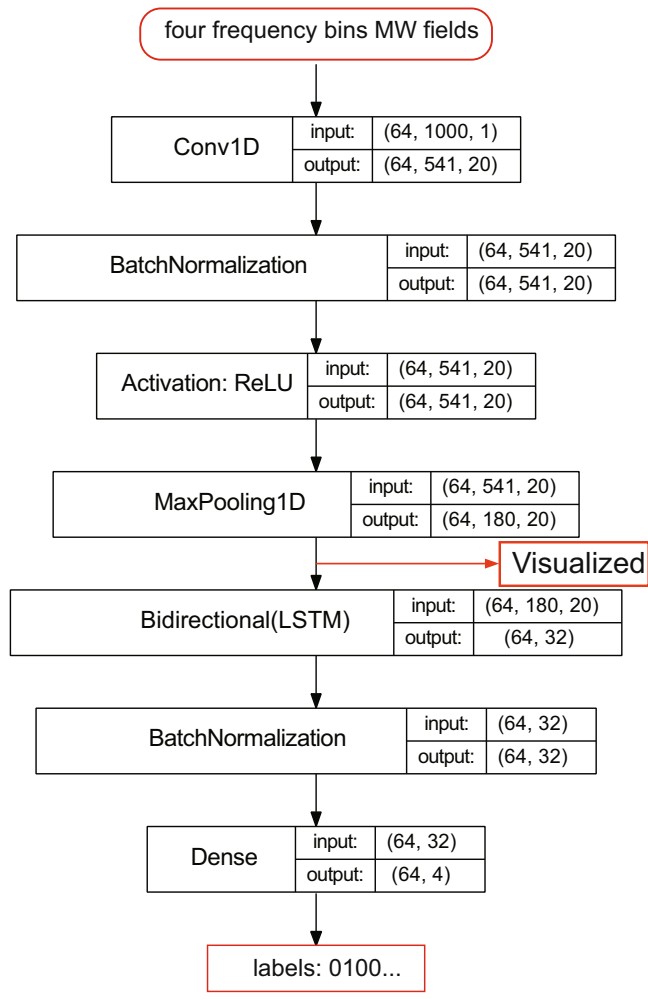

**Fig. 7 Structure of our deep learning model and size of the data.** The ReLU activation function is $f(x) = \max(0, x)$. After the data output from the max-pooling layer, the visualizations are performed; see Supplementary Materials for more details.

The network is implemented using the Keras 2.3.1 framework on Python 3.6.11 (ref. [30]). All weights are initialized with the Keras default. The hyper-parameters of the deep learning model (including the convolution kernel length, the number of hidden variables and the learning rate) are tuned using Optuna[43].

**Deep learning pipeline**. To obtain better fitting results, the data are scaled based on their maximum and minimum values, i.e., $T' = (T_i - \min(T))/(\max(T) - \min(T))$. The labels are encoded in dense vectors with four elements rather than in one-shot encoding vectors to save space[32]. Each of these elements is either 0 or 1, representing the relative phase 0 or $\pi$ of each bin, respectively.

A one-dimensional convolution layer (1D CNN), a bidirectional long–short-term memory layer (Bi-LSTM) and a dense layer are used in our deep learning model. The deep learning model structure is shown in Fig. 7. The data size for the input layer is given in the form (batch size, length of probe spectrum, number of features). The batch size is 64 in our case. Because the duration of the spectrum ranges from $t = 0$ to $t = 0.999$ ms with a time difference of $\tau = 1$ µs, the spectrum length is 1000. For a 1D input, the number of features is 1. Therefore, the data size for the input layer is (64, 1000, 1).

During training of this model, fourfold cross-validation is used to save the amount of training data. The data set is split as shown in Fig. 8. First, the data set is split into two parts. The first is the test set (red), which remains untouched during training. The second (purple) is used to train the model. In the cross-validation process, the rest data set (purple) is copied four times and is divided equally into four parts each. One of these parts is the validation data set (green) and the others are used as training sets (blue). Four models are trained on the different training sets and validation sets. Then the best model is chosen according to the validation set and is tested on the test set. After splitting, the training set, the validation set, and the test set all remain unchanged. In every epoch, each model iterates the training set only once. There is no new set being taken; instead, the same training set is iterated once each epoch.

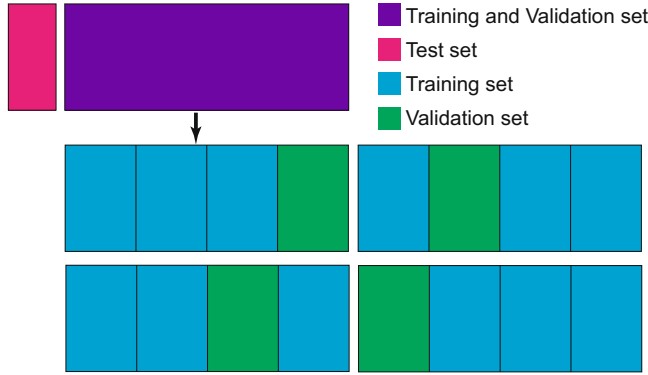

**Fig. 8 Data partition during training, validating, and testing.** First, the data are split into two sets. The first is the test set. The remaining data set is copied four times and is then split into four sets, each with different parts that act as training sets and validation sets for the training of four deep learning models.

The computational graph is cleared before each training sequence to prevent leakage of the validation data. Gaussian noise (where the mean is 0 and the standard deviation is 0.5) is added to the training data to increase the robustness of the proposed model. In addition, the learning rate is adjusted during training to jump out of the local minimum, which results in the jump in Fig. 2c in the main text. The initial learning rate is 0.001. If the loss (mean-square error) of the validation set does not decrease over 10 epochs, the learning rate is multiplied by 0.1. The RMSprop optimizer is used to update the weight of each layer during training[42].

The bidirectional LSTM layer can be replaced with the well-known self-attention layer to improve the memory of our proposed model further[44]. However, this would require more training time and increased GPU memory. The current model has been able to meet our requirements to date.

## Data availability
The data are available in Github[45] (https://github.com/ZongkaiLiu/Deep-learning-enhanced-Rydberg-multifrequency-microwave-recognition). The deep learning results are presented in the Jupyter notebook. And the master equation results are presented in the Mathematica notebooks.

## Code availability
The codes are provided in Github[45] (https://github.com/ZongkaiLiu/Deep-learning-enhanced-Rydberg-multifrequency-microwave-recognition).

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

## Acknowledgements

Z.-K.L. gratefully acknowledges the instructive discussion about deep learning with Yue Chen at the National Engineering Laboratory for Speech and Language Information Processing and the enlightenment of leraning machine learning during studying with Dr. Lei Gong at Department of Optics and Optical Engineering at USTC. D.-S.D. acknowledges funding from the National Key R&D Program of China (Grant No. 2017YFA0304800), the National Natural Science Foundation of China (Grant Nos. U20A20218, 61525504, 61435011), the Anhui Initiative in Quantum Information Technologies (Grant No. AHY020200), the Youth Innovation Promotion Association of the Chinese Academy of Sciences (Grant No. 2018490), and the major science and technology projects in Anhui Province. B.-S.S. acknowledges funding from the National Natural Science Foundation of China (Grant No. 11934013). We thank David MacDonald, MSc, from Liwen Bianji, Edanz Editing China (www.liwenbianji.cn/ac), for editing the English text of a draft of this manuscript.

## Author contributions

D.-S.D. conceived the idea for the study. Z.-K.L. conducted the physical experiments and designed the deep learning model and communication protocols. Z.-K.L. derived the theoretical formula. Z.-K.L. analysed the data with assistance from L.-H.Z., B.L., and Z.-Y.Z. The manuscript was written by Z.-K.L. The research were supervised by D.-S.D., B.-S.S. and G.-C.G. All authors contributed to discussions regarding the results and analysis contained in the manuscript.

## Competing interests

The authors declare no competing interests.
