## [Peer Review File · Nature Communications]

Deep learning enhanced Rydberg multifrequency microwave recognitionREVIEWER COMMENTS

Reviewer #1 (Remarks to the Author):

In their manuscript "Deep learning enhanced Rydberg multifrequency microwave recognition," the authors use an electromagnetically induced transparency (EIT) scheme with Rb Rydberg atoms in a hot vapor cell for detection of phase-modulated multi-frequency microwave (MW) fields. The Rydberg atoms function as microwave antennas. The microwave fields couple two nearby Rydberg states together and cause a modulation of the EIT signal. The modulated EIT signal data is analyzed via a deep learning neural network. The robustness to MW noise and the speed of the deep learning method is tested against solving the Lindblad master equation and fitting EIT signals, and shown to be superior. The authors argue that the deep learning method is superior because of the complicated interference patterns that are hard to predict and fit by solving the Lindblad master equation.

The manuscript is well-written and clear. I applaud the authors for including very good illustrations of the setup and methods, Figs. 1 and 2, which are immensely helpful. The use of Rydberg EIT for microwave sensing has gained broad interest in the atomic physics community, especially in recent years due to the National Quantum Initiative and the related focus on quantum technologies such as quantum-enhanced sensing. Therefore, I think the topic is timely. As far as I know, the application of a deep learning method to Rydberg EIT signals is novel. In my opinion, the ideas and results presented here are interesting and will be of interest to the broader community as well.

I think there are two important results in this manuscript. The first result is the demonstration that Rydberg EIT can be used to sense multi-frequency MW fields in the presence of noise, which is important in my opinion. The second result is that deep learning is a powerful method to extract the message from the complicated interference patterns in the EIT signal.

The results presented in the manuscript are interesting and warrant publication in Nature Communications, in my opinion.

However, I have a three issues with the current manuscript that I think should be addressed before it can be published:

1.) On page 5, the authors claim "..., the prediction time for the master equation is 25 s per spectrum, while the time for the deep learning method is 1.6 ms per spectrum." I think this claim is meaningless unless the authors specify what programming language was used, and what computer was used. For example, if the deep learning code runs on a GPU, and the master equation code runs on a CPU, how

can absolute execution times like this be compared? Maybe the master equation code was written in a different programming language than the deep-learning code, and the language is just slower to execute. I think for a time comparison, it must be ensured the code runs on the same machine the same processor, and is written in the same language. At the very least, the authors should clearly state here what computers and programming languages are used to make this comparison.

2.) The authors frequently mention MW noise, but what is the source and nature of the noise? Is it noise that the authors add to the MW signal on purpose? If yes, how is it added? Is it electrical noise generated by the MW source which cannot be avoided? Is it pick-up noise from some background radiation present in the lab (cellphone towers etc.)? Is it white noise? What is the spectrum? Does the noise change over time, maybe it is less during night time? I think that more information on the noise should be added to the manuscript.

3.) On page 5, the authors state "This implies that when more MW bins are used, the interference term cannot be ignored and the master equation must be adjusted to increase its accuracy". I think this requires a bit more explanation in the manuscript. Did the authors try to add this interference term? Why or why not? I think a sentence or two should be added here or in the methods section about this interference term, and why it was or was not implemented.

In addition to the three bigger points above, I found a small grammatical issue that should be corrected:

4.) Second sentence: "For the exaggerated..." should be changed to "Due to the exaggerated..."

Reviewer #2 (Remarks to the Author):

The manuscript by Liu and colleagues describes the neural-network-supported analysis of transmission spectra. These emerge from laser-microwave interaction in a gas of 85Rb atoms at electromagnetically-induced-transparency (EIT) conditions involving highly-excited Rydberg states. Information is encoded in the relative phases of different microwave frequencies, which thus maps onto the atomic absorption spectrum. A neural network is trained to analyze and extract the information.

I appreciate that the individual parts of Rydberg EIT, on the one hand, and implementation of the neural network, on the other hand, are decent pieces of work. My overall assessment of this manuscript is,

however, that I find it not well suited for Nature Communications. My main criticism is that the neural network, in all regards, is a black box, and it is used in a very similar way as many cases before. The presented work does not help to understand the microscopic mechanisms of Rydberg EIT, the noise sources (in fact, noise is never shown, quantified, or modified in the manuscript), or the limits of performance further. The use of the neural network boils down to an application used in many atom-physics labs worldwide to optimize some kind of experimental outcome, such as quantum gas production or pulse optimization. Therefore, the specific application might be new, but I do not consider the approach to be a new contribution nor yielding any novel physics insight. I believe the manuscript to be better suited for a technical or instrumentational journal, where the details of the neural network presented can be better appreciated.

But even in such a more specialized journal, I believe some issues need to be improved. I want to give a few points which the authors might want to consider.

- There is a strong imbalance between discussing the neural network and the physical system. For the neural network, we learn the detailed structure of each layer, repeated in the caption of Fig. 1 and in the text. By contrast for the physical implementation, we don't learn about the atomic density (necessary for the optical density), there is no absorption spectrum shown (the reader cannot estimate the signal-to-noise ratio), and despite the claim that the system is superior in dealing with or rejecting noise, we don't get a quantitative value for the noise level, nor a systematic assessment of the performance depending on the noise source.
- For the discussion in a more specialized journal, I suggest bringing the details of the neural network from Fig. 2 (c)-(g) into a new figure which can be referenced in the "deep learning" section.
- I suggest showing an absorption spectrum in Fig. 1 and a microwave spectrum without (and possibly with noise – if analyzed) in Fig. 2. In any case, the noise discussion must rise to a quantitative level to substantiate the claimed neural-network advantage.
- I was confused by the information on the training data in the caption of Fig. 2. From the text (page 3, left column), I understand that the measured data takes the form of probe spectrum $\{T\}$ and phase information $\{\phi_1, \phi_2, \phi_3, \phi_4\}$. In the caption, however, the training data is given as, e.g., 131x4, "where 4 comes from four-fold cross validation". Please clarify; I don't understand the meaning of these numbers.
- It is unclear to me how an epoche is defined. Again, from page 3, left column, I understand that the data set is distributed to training, validation, and test sets. But it is unclear if in every epoche, a new set is taken. Please clarify.

Reviewer #3 (Remarks to the Author):

Dear editor,

the authors present an interesting setup, where Rydberg atoms act as a receiver, whose performance is enhanced by a deep learning model. The latter manages to extract the binary data with high fidelity. The authors compare their approach with a direct simulation via the Lindblad master equation. I found the work quite interesting and I am, in principle, positive for publication in Nature Communications. However, I cannot fully judge the importance of the work based on the presented evidence.

What I am missing most is a benchmark with other mw detectors. For instance, how sensitive is the antenna? How do classical mw receivers perform with respect to their own work? How does the phase encoding perform with other types of encoding? A discussion of these points would not only put the results in a more general perspective regarding the performance, it would also make the results more comprehensive for readers outside the field of Rydberg atoms.

My second point concerns the comparison with the master equation. First, I would prefer to see somewhere a plot of a typical measurement signal together with a simulation to get an idea of the amount of noise. Second, I would expect a discussion of the noise sources of the system (atomic collisions, decoherence, loss, dephasing) and the influence on the signal. What do the authors expect? Are these noise sources representable with the Lindblad master equation, and how? Finally, I would also like to request a discussion of alternative approaches to extract the information from the signal. What about effective fit functions, etc,...?

Apart from that the paper is well written and understandable.

REVIEWER COMMENTS

Reviewer #1 (Remarks to the Author):

Reviewer Comment:

In their manuscript "Deep learning enhanced Rydberg multifrequency microwave recognition," the authors use an electromagnetically induced transparency (EIT) scheme with Rb Rydberg atoms in a hot vapor cell for detection of phase-modulated multi-frequency microwave (MW) fields. The Rydberg atoms function as microwave antennas. The microwave fields couple two nearby Rydberg states together and cause a modulation of the EIT signal. The modulated EIT signal data is analyzed via a deep learning neural network. The robustness to MW noise and the speed of the deep learning method is tested against solving the Lindblad master equation and fitting EIT signals, and shown to be superior. The authors argue that the deep learning method is superior because of the complicated interference patterns that are hard to predict and fit by solving the Lindblad master equation.

The manuscript is well-written and clear. I applaud the authors for including very good illustrations of the setup and methods, Figs. 1 and 2, which are immensely helpful. The use of Rydberg EIT for microwave sensing has gained broad interest in the atomic physics community, especially in recent years due to the National Quantum Initiative and the related focus on quantum technologies such as quantum-enhanced sensing. Therefore, I think the topic is timely. As far as I know, the application of a deep learning method to Rydberg EIT signals is novel. In my opinion, the ideas and results presented here are interesting and will be of interest to the broader community as well.

I think there are two important results in this manuscript. The first result is the demonstration that Rydberg EIT can be used to sense multi-frequency MW fields in the presence of noise, which is important in my opinion. The second result is that deep learning is a powerful method to extract the message from the complicated interference patterns in the EIT signal.

Reply:

Thank you for your very positive comments on our manuscript. Our work using deep learning in a Rydberg system has shown enhanced performance in multi-frequency MW field sensing and communication with noise, which we believe will encourage the application of Rydberg systems in more complex environments.

Reviewer Comment:

The results presented in the manuscript are interesting and warrant publication in Nature Communications, in my opinion.

However, I have three issues with the current manuscript that I think should be addressed before it can be published:

1.) On page 5, the authors claim "..., the prediction time for the master equation is 25 s per spectrum, while the time for the deep learning method is 1.6 ms per spectrum." I think this claim is meaningless

unless the authors specify what programming language was used, and what computer was used. For example, if the deep learning code runs on a GPU, and the master equation code runs on a CPU, how can absolute execution times like this be compared? Maybe the master equation code was written in a different programming language than the deep-learning code, and the language is just slower to execute. I think for a time comparison, it must be ensured the code runs on the same machine the same processor, and is written in the same language. At the very least, the authors should clearly state here what computers and programming languages are used to make this comparison.

Reply:

Thank you for your comment, which raises a good point. The deep learning code and the master equation code do run on the same computer but were written in different programming languages. The deep learning code was written in Python 3.7.6, while the master equation code was written in Mathematica 11.1. Even if they were to be written in the same language (e.g., both written in Python), the packages to be imported would be different. The optimization process requirements for the hardware and calculation resources are also different. It is thus unlikely that we could compare the running times under the same conditions. Additionally, a CPU is good at performing complex calculations (i.e., solving the master equation) with few cores, while a GPU is faster when performing simple tasks (i.e., deep learning) using many more cores than the CPU. Therefore, we have only added the type of hardware and the programming languages used to the revised manuscript and have not emphasized the high speed of the deep learning model.

Manuscript changes:

- **Section “Comparison between deep learning method and the master equation”:** The master equation is solved by “FindFit” function in Mathematica 11.1 with both “AccuracyGoal” and “PrecisionGoal” default, while the deep learning code is written in Python 3.7.6. These codes are run on the same computer with NVIDIA GTX 1650 and Intel® Core™ i7-9750H.
- **Section “Comparison between deep learning method and the master equation”:** The sentence “The deep learning model's response speed is much faster than the master equation fitting process” has been removed.
- **Conclusion:** The sentence “Furthermore, the blend of Rydberg atoms and deep learning reduces the response time required to receive and decode messages” has been removed.

Reviewer Comment:

2.) The authors frequently mention MW noise, but what is the source and nature of the noise? Is it noise that the authors add to the MW signal on purpose? If yes, how is it added? Is it electrical noise generated by the MW source which cannot be avoided? Is it pick-up noise from some background radiation present in the lab (cellphone towers etc.)? Is it white noise? What is the spectrum? Does the noise change over time, maybe it is less during nighttime? I think that more information on the noise should be added to the manuscript.

Reply:

Thank you for your comment. We will address your questions in the following.

1. The “noise” considered here refers to a combination of additional noise and systematic noise. In this case, we call the noise that comes from atoms and the external environment “systematic noise” and call the noise that was added on purpose “additional noise”. The systematic noise cannot be adjusted quantitatively, and we will discuss it with its noise spectrum later. We added the additional noise on purpose to test the robustness of both our deep learning model and the master fitting method to the noise. The additional noise is white noise (which is common in reality) with a mean μ and a standard deviation σ . The transmitted signals after the additional noise were added are shown in Figure C1. As the standard deviation σ increasing, the signal is hard to fit by the master equation. We ignore the $1/f$ noise in this case because it decays rapidly in the low-frequency range and the signal with which it would interfere is located within the 2–200 kHz range.
2. Because the noise on the data set is independent and is distributed identically (the entire data set is shuffled before being split into the training and test sets), the systematic noise pattern is almost the same in both the training set and the test set. **The deep learning model has already learned the systematic noise pattern during the training process, which is one of the major advantages of deep learning against systematic noise.** Based on that knowledge, we added the additional noise to the training set to increase the robustness of the model to the “unseen” noise and tested its robustness by adding the additional noise to the test set. The results of this process are shown in Figure C2 and a comparison between the results obtained from the deep learning method and those from the master equation fitting is shown in Figure C3.
3. However, there is a case where the noise is not independent or identically distributed (i.e., the case where a specific noise occurs during testing only). This problem can be solved by online learning and addition of prior knowledge as new features into the data, e.g., data for the temperature, the weather, and other factors [arXiv:1711.03705 [cs.LG]]. However, this is beyond the scope of the present work.
4. With regard to the sources of the systematic noise in our experiment, the noise comes from two sides: the first type is from the exterior, such as the noise sources that you mentioned above; the second type is from the interior (e.g., from atoms in the vapor cell), which was noted by Reviewer #3. As stated in the literature [Nat. Phys. 16, 911–915 (2020)], there are different types of noise for different frequencies. For frequencies below 1 kHz, there is $1/f$ noise. Then, for higher frequencies (in the 1 kHz–100 kHz range), there is noise from atomic transitions outside and inside the light area. At much higher frequencies (e.g., in the 100 kHz–1 MHz range), the noise comes from the control systems of the lasers. During the experiments, we found that our deep learning model was robust with respect to these noise types.

Figure C1. Fitting curve for the master equation on data with additional white noise. The standard deviation values of the white noise are $\sigma = 0$ (a), 0.05 (b), 0.55 (c), 1.05 (d), 1.55 (e), and 2.05 (f), respectively. The prediction results are $(0, 0, \pi, 0)$, $(0, 0, \pi, 0)$, $(0, 0, \pi, 0)$, $(0, 0, \pi, 0)$, $(0, 0, \pi, 0)$, and $(0, 0, 0, 0)$, while the ground truths are both $(0, 0, \pi, 0)$.

Figure C2. Deep learning model accuracy on the noisy test set after training on the noisy training set. The x - and y -axes represent the standard deviations of the additional white noise added to the test set and the training set, respectively. The color bar represents the accuracy of the model when used on the noisy test set. The results were obtained by averaging five sets of predictions performed using the deep learning model. The diagonal (red line) indicates the accuracy of the model on a test set in which the noise distribution is the same as that of the training set. Figure C2 also demonstrates the performance of the deep learning model when used on a data set with biased or unbalanced noise. The results below the red line show the performance of the model after training on a weaker-noise training set when predicting based on a stronger-noise test set, i.e., generalization for a stronger noise case. These results indicate that the deep learning model has the ability of generalization required to adapt to stronger noise. In the area above the red line, there is more noise in the training set than in the test set. Theoretically, a small amount of additional noise in the training set will increase the robustness of the deep learning model. However, when the noise increases, it affects the accuracy, which decays rapidly.

Figure C3. Prediction accuracy of the deep learning model and the master equation on the noisy test set, where the noise is white noise with mean $\mu = 0$ and a standard deviation σ . The noise is added quantitatively using different σ values, as suggested by Reviewer #2. Before and after addition of the noise, the data are scaled between 0 and 1 using their maximum and minimum. The deep learning model is trained in a training set without additional noise. In addition, we do not involve noise spectrum when solving the master equation. Each point is obtained after averaging of five predictions based on the noisy data. Both prediction accuracy curves drop as the standard deviation of the white noise increases. However, the deep learning method always outperforms the master equation approach.

Figure C4. Systematic noise spectra. The blue curve is the spectrum measured when the differential photodetector was switched off. The red curve is the spectrum measured when the detector was switched on but without a light signal. The yellow curve is the spectrum measured with the EIT configuration but without the microwave signal. The resolution bandwidth and the video bandwidth of the spectrum analyzer (Ceyear 4024F) are both 10 Hz, and the attenuation is 10.

Manuscript changes:

- We add Fig. 2(e) in the main text to show the robustness to the noises of the deep learning model when adding the unbalanced noise in the training set and test set. The caption is changed. The discussion of Fig. 2(e) is added in the paragraph above section “Comparison between deep learning method and the master equation”.
- Figure 4 in the main text is also added to compare the robustness of the deep learning model and the master equation. The discussion of Fig. 4 is added in the first paragraph in section “Comparison between deep learning method and the master equation”.
- The spectrum of the systematic noise and the transmission with additional noise are shown in supplementary materials.
- **Methods section, “Generation and calibration of MW fields.”:** The type of spectrum

analyzer has been given.

Reviewer Comment:

3.) On page 5, the authors state "This implies that when more MW bins are used, the interference term cannot be ignored and the master equation must be adjusted to increase its accuracy". I think this requires a bit more explanation in the manuscript. Did the authors try to add this interference term? Why or why not? I think a sentence or two should be added here or in the Methods section about this interference term, and why it was or was not implemented.

Reply:

Thank you for this comment. After reconsideration, we realized that this was an error in the previous manuscript. The failure of the master equation to fit the transmission spectrum for larger numbers of MW bins is the result of an **inherent defect in the fitting method** rather than the effect of the interference term. Specifically, in the case of 20 bins, there is a disadvantage that comes from the fitting method itself, i.e., it can easily become trapped by local minima. Some type of prior knowledge is required to overcome this disadvantage, e.g., provision of the initial values of the phases and limitation of some values before fitting. In contrast, the deep learning model is data-driven and does not require any prior knowledge. The local minima problem of deep learning can be overcome using some well-known techniques, including learning rate scheduling and the design of a more effective optimizer [ISBN: 978-1-61729-443-3].

Manuscript changes:

- **Section “Comparison between deep learning method and the master equation”:** This implies that there is a disadvantage that comes from the fitting method itself, i.e., it can easily become trapped by local minima. Some type of prior knowledge is required to overcome this disadvantage, e.g., provision of the initial values of the phases and limitations of some values before fitting. In contrast, the deep learning model is data-driven and does not require any prior knowledge. The local minima problem of deep learning can be overcome using some well-known techniques, including learning rate scheduling and the design of a more effective optimizer [33].
- **Section “Comparison between deep learning method and the master equation.”:** The sentence “In our case, the effect of interference between the MW bins that results from increasing the number of bins on the fitting accuracy is greater than that of the increased noise that results from increasing the DD bandwidth for high-speed signals” has been replaced with “In our case, the impact of the number of bins is greater than increasing the DD bandwidth for high-speed signals on the fitting accuracy.”

Reviewer Comment:

In addition to the three bigger points above, I found a small grammatical issue that should be corrected:

4.) Second sentence: "For the exaggerated..." should be changed to "Due to the exaggerated..."

Reply:

Thank you for pointing this out. This sentence has been amended accordingly in the revised version of the manuscript.

Manuscript change:

- **Abstract:** “Due to the exaggerated properties of Rydberg atoms”

Reviewer #2 (Remarks to the Author):

Reviewer Comment:

The manuscript by Liu and colleagues describes the neural-network-supported analysis of transmission spectra. These emerge from laser-microwave interaction in a gas of 85Rb atoms at electromagnetically-induced-transparency (EIT) conditions involving highly-excited Rydberg states. Information is encoded in the relative phases of different microwave frequencies, which thus maps onto the atomic absorption spectrum. A neural network is trained to analyze and extract the information.

Reply: Thank you for your comment. This is an accurate summary of the work.

Reviewer Comment:

I appreciate that the individual parts of Rydberg EIT, on the one hand, and implementation of the neural network, on the other hand, are decent pieces of work. My overall assessment of this manuscript is, however, that I find it not well suited for Nature Communications. My main criticism is that the neural network, in all regards, is a black box,

Reply:

Thank you for your comment. Recently, there have been some explorations of the interpretability of neural networks and attempts have been made to open this “black box”, including visualization of the convolution kernel of a CNN, the hidden layers of an RNN, and the attention mechanisms [ISBN: 978-1-61729-443-3; DOI: 10.1109/VAST.2017.8585721; arXiv:1908.04626 [cs.CL]]. In addition, deep learning and artificial intelligence have recently provided significant advantages in many fields, including protein structure prediction [Nature 577, 706–710 (2020); Nature 596, 583–589 (2021); DOI: 10.1126/science.abj8754; Nature 596, 590–596 (2021)], high-performance brain-to-text communication via handwriting [Nature 593, 249–254 (2021)], vortex light recognition [Phys. Rev. Lett. 123, 183902 (2019); Phys. Rev. Lett. 124, 160401 (2020)], and phase recognition in physics [DOI: 10.1038/NPHYS4037; DOI: 10.1038/NPHYS4035]. These methods extract the features and correlation properties of massive data without either human intervention or prior knowledge. These works help to accelerate the pace of scientific discovery itself and thus explore the boundaries of science.

Our experiment using deep learning in a Rydberg system shows **enhanced performance in sensing multi-frequency microwave electric fields**. This experiment promotes the possibility of applications of the Rydberg system in more complex environments.

Manuscript change:

- **Introduction:** add reference vortex light recognition [Phys. Rev. Lett. 123, 183902 (2019), Phys. Rev. Lett. 124, 160401 (2020)].

Reviewer Comment:

and it is used in a very similar way as many cases before. The presented work does not help to understand the microscopic mechanisms of Rydberg EIT, the noise sources (in fact, noise is never

shown, quantified, or modified in the manuscript), or the limits of performance further. The use of the neural network boils down to an application used in many atom-physics labs worldwide to optimize some kind of experimental outcome, such as quantum gas production or pulse optimization.

Reply:

Thank you for your comment. In the manuscript, we emphasize that we use the deep learning model in the quantum sensing field to suppress the noise-induced effects by the high sensitivity of the Rydberg atoms, which is significantly distinct from applications such as quantum gas production or pulse optimization. This deep learning model **turns this side effect into an advantage**, showing a similar effect to that reported in the literature [Phys. Rev. X 10, 031029 (2020)], but this work is in the quantum sensing field. Without human intervention and prior knowledge (e.g., the master equation), the deep learning model performs well after few-sample training.

Furthermore, the deep learning-enhanced Rydberg system helps us to deal with interactions with multiple MW electric fields, i.e., it reduces the impact of the noise and distortion to the input MW bins when there are more than 20 MW bins, which is promising for use in studies of the channel capacity limitations. Because spectra that are difficult for humans to recognize as a result of noise and distortion are distinguishable when using the deep learning model, **Rydberg systems enhanced by deep learning could take steps toward the realization of the capacity limit** proposed in the literature [Phys. Rev. Lett. 121, 110502 (2018)].

Manuscript change:

- **The paragraph above Conclusion:** “Besides the transmission rate, this deep learning enhanced Rydberg system promises for use in studies of the channel capacity limitations. Because spectra that are difficult for humans to recognize as a result of noise and distortion are distinguishable when using the deep learning model, Rydberg systems enhanced by deep learning could take steps toward the realization of the capacity limit proposed in the literature Ref.[35].”

Reviewer Comment:

Therefore, the specific application might be new, but I do not consider the approach to be a new contribution nor yielding any novel physics insight.

Reply:

Thank you for your comment. As mentioned by Reviewer # 1 (“The use of Rydberg EIT for microwave sensing has gained broad interest in the atomic physics community”), new findings in Rydberg EIT for microwave sensing are likely to be of interest to the atomic physics community. We believe that our results contain a new physics that comprises deep learning and atomic technology and shows enhanced quantum sensing. Deep learning could enable us to advance the frontiers of physics beyond the previous limits of human knowledge, including aspects such as the master equation and Rydberg MW sensing.

Reviewer Comment:

I believe the manuscript to be better suited for a technical or instrumentational journal, where the

details of the neural network presented can be better appreciated.

But even in such a more specialized journal, I believe some issues need to be improved. I want to give a few points which the authors might want to consider.

- There is a strong imbalance between discussing the neural network and the physical system. For the neural network, we learn the detailed structure of each layer, repeated in the caption of Fig. 1 and in the text. By contrast for the physical implementation, we don't learn about the atomic density (necessary for the optical density), there is no absorption spectrum shown (the reader cannot estimate the signal-to-noise ratio), and despite the claim that the system is superior in dealing with or rejecting noise, we don't get a quantitative value for the noise level, nor a systematic assessment of the performance depending on the noise source.

Reply:

Thank you for your helpful comments, which we have addressed as follows.

1. The temperature and the atomic density have been added to the caption of Fig. 1.
2. The absorption spectra are included in the supplementary materials.
3. The additional noise is quantified using its standard deviation σ .
4. The spectrum of the systematic noise has been added to the supplementary materials.

The noise includes both the additional noise and the systematic noise. In this case, we name the noise from atoms and the external environment “systematic noise”, while the noise that is added deliberately is called “additional noise”. Please see the reply to point #2 for Reviewer #1 to get more about the noise analysis.

Manuscript changes:

- **Fig. 1 caption:** added temperature and atomic density calculated by Python package ARC [Computer Physics Communications 220, 319–331 (2017)].
- The spectrum and the master equation fitting results are included in supplementary materials as Fig. S1.
- The performance of the deep learning model on the additional noise quantified by the standard deviation is added in Fig. 2(e)
- The comparison between the deep learning model and the master equation fitting results has been included in Fig. 4.
- The spectrum of systematic noise is added in supplementary materials as Fig. S2.

Reviewer Comment:

- For the discussion in a more specialized journal, I suggest bringing the details of the neural network from Fig. 2 (c)-(g) into a new figure which can be referenced in the “deep learning” section.

Reply:

Thank you for your comment. We assume that you intended to mention Fig. 1 here, because Fig. 2 does not have panels (f) and (g). We agree that Fig. 1(f) and (g) may be too complex for readers without a deep learning background and have therefore moved these figures to the Methods section.

However, Figs. 1(c)–(e) serve to provide brief introductions to the deep learning layers and therefore remain as part of the main text.

Manuscript change:

- Fig. 1(f), (g) have been moved to the Methods section as Fig. 6.

• I suggest showing an absorption spectrum in Fig. 1 and a microwave spectrum without (and possibly with noise – if analyzed) in Fig. 2. In any case, the noise discussion must rise to a quantitative level to substantiate the claimed neural-network advantage.

Reply:

Thank you for your comment, which we have addressed as follows.

1. We have added probe transmission data with and without noise to the supplementary materials.
2. The systematic noise spectrum is presented in the supplementary materials
3. As also suggested by Reviewer #1, Fig. 2(e) has been added to illustrate the robustness of the deep learning method, where the additional noise is white noise and is quantified using its standard deviation σ .
4. Fig. 4 has been added to compare the fitting performances of the deep learning model and the master equation on noisy data sets.

Please see the reply to point #2 for Reviewer #1 to get more about the noise discussion.

Manuscript changes:

- The spectrum and the master equation fitting results are added in supplementary materials, as Fig. S1.
- The performance of the deep learning model on the additional noise quantified by the standard deviation is included in Fig. 2(e)
- The comparison between the deep learning model and the master equation fitting results has been added in Fig. 4.
- The systematic noise spectrum is presented in supplementary materials, as Fig. S2.

Reviewer Comment:

• I was confused by the information on the training data in the caption of Fig. 2. From the text (page 3, left column), I understand that the measured data takes the form of probe spectrum $\{T\}$ and phase information $\{\phi_1, \phi_2, \phi_3, \phi_4\}$. In the caption, however, the training data is given as, e.g., 131x4, “where 4 comes from four-fold cross validation”. Please clarify; I don’t understand the meaning of these numbers.

Reply:

Thank you for your comment. Cross-validation is a trick used in deep learning to enable better training of the model. The data set is split as shown in Figure C5 below. First, the data set is split into two parts. The first is the test set (red), which remains untouched during training. The second (purple) is used to train the model. In the cross-validation process, the second data set (purple) is copied four times and each set is then divided equally into four parts. One of these four parts is used as the validation data set (green), and the others are used as the training sets (blue). Four models are

trained on the different training sets and validation sets. The best model is then selected based on the validation set and is tested on the test set. Therefore, the quoted numbers should be “ 131×3 ” rather than “ 131×4 ”. To avoid any confusion, this has been changed to the total size of the training set, i.e., 393.

Figure C5. First, the data are split into two sets. The first is the test set. The remaining data set is copied four times and is then split into four sets, each with different parts that act as training sets and validation sets for the training of four deep learning models.

Manuscript changes:

- **Fig. 2. Caption:** “The training data numbers are (b) 393 and (c) 1194. The validation data numbers are (b) 131 and (c) 398; more details about the data set split are presented in the Methods section.”
- **Methods section “Deep learning pipeline”:** “The data set is split as shown in Fig. 8. First, the data set is split into two parts. The first is the test set (red), which remains untouched during training. The second (purple) is used to train the model. In the cross-validation process, the rest data (purple) is copied four times and is divided equally into four parts each. One of these parts is the validation data set (green) and the others are used as training sets (blue). Four models are trained on the different training sets and validation sets. Then the best model is chosen according to the validation set and is tested on the test set.”
- Fig. 8 has been added.

Reviewer Comment:

• It is unclear to me how an epoch is defined. Again, from page 3, left column, I understand that the data set is distributed to training, validation, and test sets. But it is unclear if in every epoch, a new set is taken. Please clarify.

Reply:

Thank you for your comment. The initial data set is split as shown in Figure C5. Please refer to the

last apply. After splitting, the training set, the validation set, and the test set all remain unchanged. In every epoch, each model iterates the training set only once. There is no new set being taken; instead, **the same** training set is iterated once each epoch.

Manuscript change:

- **Methods section, “Deep learning pipeline”:** “After splitting, the training set, the validation set, and the test set all remain unchanged. In every epoch, each model iterates the training set only once. There is no new set being taken; instead, the same training set is iterated once each epoch.”

Reviewer #3 (Remarks to the Author):

Dear editor,

Reviewer Comment:

the authors present an interesting setup, where Rydberg atoms act as a receiver, whose performance is enhanced by a deep learning model. The latter manages to extract the binary data with high fidelity. The authors compare their approach with a direct simulation via the Lindblad master equation.

Reply: Thank you for your comment. We believe that it represents an accurate summary of the work.

Reviewer Comment:

I found the work quite interesting and I am, in principle, positive for publication in Nature Communications.

Reply: Thank you for this positive assessment of our work.

Reviewer Comment:

However, I cannot fully judge the importance of the work based on the presented evidence. What I am missing most is a benchmark with other mw detectors. For instance, how sensitive is the antenna? How do classical mw receivers perform with respect to their own work? How does the phase encoding perform with other types of encoding? A discussion of these points would not only put the results in a more general perspective regarding the performance, it would also make the results more comprehensive for readers outside the field of Rydberg atoms.

Reply:

Thank you for your comments, which we will address in the following.

1. One of the advantages of the Rydberg atom is that the accuracy of the Rydberg atom approaches the photon shot noise limit [Opt. Express 25, 8625–8637 (2017)]. In principle, the accuracy of the Rydberg atom is higher than that of the classical antenna. According to recent work based on the atomic superheterodyne method [Nature Physics 16, 911–915 (2020)], ultrahigh sensitivity can be obtained. However, in this proof-of-principle demonstration, there is considerable room for the optimization required to reach that limit (e.g., stabilization of the laser, narrowing the laser linewidth, and temperature stabilization).
2. In classical MW receivers, the wavelength of the received MW is limited by the size of the antenna. In contrast, for Rydberg atoms to receive MWs with different wavelengths, the only part of the system that needs to be tuned is the frequency of the laser [DOI: 10.1109/TAP.2014.2360208].
3. Other encoding types, e.g., frequency division multiplexing amplitude shift keying (FDM-ASK) and frequency division multiplexing quadrature amplitude modulation (FDM-QAM), can be also decoded using the deep learning model. The decoding processes for these two types of signals are like that for FDM-PSK. Therefore, we only show the FDM-PSK decoding process for demonstration purposes.

Manuscript changes:

- **Conclusion:** “The frequency of carrier to be decoded covers from several hertz to terahertz since for Rydberg atoms to receive MW with different wavelengths, the only part of the system that needs to be tuned is the frequency of the laser, while in classical receivers, the wavelength of the received MW is limited by the size of the antenna [37-40].”
- **Conclusion:** “One of the advantages of the Rydberg atom is that the accuracy of the Rydberg atom approaches the photon shot noise limit [35]. In principle, the accuracy of the Rydberg atom is higher than that of the classical antenna. According to recent work based on the atomic superheterodyne method, ultrahigh sensitivity can be obtained [10]. However, in this proof-of-principle demonstration, there is considerable room for the optimization required to reach that limit (e.g., stabilization of the laser, narrowing the laser linewidth, and temperature stabilization).”
- **Conclusion:** “This decoding ability can be generalized further to decode other signals that are encoded by different protocols, e.g., frequency division multiplexing amplitude shift keying (FDM-ASK), frequency division multiplexing quadrature amplitude modulation (FDM-QAM), and IEEE 802.11ac WLAN standard signals for a 5 GHz carrier.”

Reviewer Comment:

My second point concerns the comparison with the master equation. First, I would prefer to see somewhere a plot of a typical measurement signal together with a simulation to get an idea of the amount of noise. Second, I would expect a discussion of the noise sources of the system (atomic collisions, decoherence, loss, dephasing) and the influence on the signal. What do the authors expect? Are these noise sources representable with the Lindblad master equation, and how? Finally, I would also like to request a discussion of alternative approaches to extract the information from the signal. What about effective fit functions, etc.,...?

Reply:

Thank you for your comment. The “noise” in this case refers to a combination of the additional noise and the systematic noise. We named the noise from atoms and the external environment “systematic noise” and called the noise that was added deliberately the “additional noise”.

1. The measured signal (black) and the master equation fitting results (red) are shown in Figure C6. Additional noise is added with standard deviations of $\sigma = 0, 0.05,$ and $1.05,$ respectively. When this additional noise is large, the master equation cannot fit the signal correctly.

2. The systematic noise spectra (Figure C7) are shown in the supplementary materials as Fig. S1. As stated in the literature [Nat. Phys. 16, 911–915 (2020)], there are different noise types for different frequencies. For frequencies below 1 kHz, there is $1/f$ noise. Then, for a higher frequency range (1 kHz–100 kHz), there is noise from atomic transitions out from and into the light area. At much higher frequencies (100 kHz–1 MHz), the noise comes from the control systems of the lasers. During the experiments, we found that our deep learning model was robust with respect to these noise types. The noise sources that you mentioned (atomic collisions, decoherence, loss, dephasing) are not major factors because, after the addition of these noises into the master equation, the transmission spectrum is still smooth. To simulate the noise rigorously, the noise spectrum must therefore be taken into the master equation. This then brings the problem that the noise is stochastic

and changes over time, which results in a large fitting error. However, the deep learning model extracts the information from the transmission spectrum after it learns the signal pattern from large quantities of data and thus achieves high accuracy. The robustness to the noise of deep learning and master equation is shown in Figure C8.

Additionally, for multiple bins, the fitting method itself has a disadvantage, i.e., it will easily become trapped by the local minima. Overcoming this disadvantage requires some type of prior knowledge, e.g., provision of the initial values of the phases and limitations of some values before fitting. In contrast, the deep learning model is data-driven and does not need this prior knowledge. The local minima problem of deep learning can be overcome using some well-known techniques, e.g., learning rate scheduling and changing the optimizer [ISBN: 978-1-61729-443-3].

3. Another method to decode the signal is available that uses an in-phase and quadrature (I-Q) demodulator or a lock-in amplifier [10.1109/LAWP.2019.2931450; Applied Physics Letters 112, 211108 (2018)]. However, the carrier frequency must be given when decoding the signal in this case. Additionally, for multiple MW bins, numerous bandpass filters are required. The deep learning method is thus much more convenient.

Figure C6. Fitting curve for the master equation on data with additional white noise. The standard deviation values of the white noise are $\sigma = 0$ (a), 0.05 (b), 0.55 (c), 1.05 (d), 1.55 (e), and 2.05 (f), respectively. The prediction results are $(0, 0, \pi, 0)$, $(0, 0, \pi, 0)$, $(0, 0, \pi, 0)$, $(0, 0, \pi, 0)$, $(0, 0, \pi, 0)$, and $(0, 0, 0, 0)$, while the ground truths are both $(0, 0, \pi, 0)$.

Figure C7. Spectra of the systematic noise. The blue curve represents the spectrum obtained when the differential photodetector was switched off. The red curve represents the spectrum acquired with the detector switched on but without the light signal. The yellow curve represents the spectrum obtained with the EIT configuration but without the microwave signal. The resolution bandwidth and the video bandwidth of the spectrum analyzer (Ceyear 4024F) are both 10 Hz, and the attenuation is 10.

Figure C8. Prediction accuracy results for the deep learning model and the master equation on the noisy test set, where the noise is white noise with mean $\mu = 0$ and a standard deviation σ . This noise is added quantitatively with different values of σ , as suggested by Reviewer #2. Before and after the addition of the noise, the data were scaled between 0 and 1 using their maximum and minimum. The deep learning model is trained using a training set without additional noise. Additionally, we do not involve noise spectrum during the process of solving the master equation. Each point is obtained after averaging five predictions based on the noisy data. Both prediction accuracy curves drop when the standard deviation of the white noise increases. However, the deep learning method always outperforms the master equation.

Manuscript changes:

- The transmission spectrum, the fitting curves, and the noise spectrum have been added to the supplementary materials.
- **Section “Comparison between deep learning method and the master equation”:** “This implies that there is a disadvantage that comes from the fitting method itself, i.e., it can easily become trapped by local minima. Some type of prior knowledge is required to overcome this disadvantage, e.g., provision of the initial values of the phases and limitations of some values before fitting. In contrast, the deep learning model is data-driven and does not require any prior knowledge. The local minima problem of deep learning can be overcome using some well-known techniques, including learning rate scheduling and design of a more effective optimizer [33].”
- **Section “Comparison between deep learning method and the master equation”:** “Another method to decode the signal is available that uses an in-phase and quadrature (I-Q) demodulator or a lock-in amplifier [7, 12]. However, the carrier frequency must be given when decoding the signal in this case. Additionally, for multiple MW bins, numerous bandpass filters are required. The deep learning method is thus much more convenient.”

Reviewer Comment:

Apart from that the paper is well written and understandable.

Thank you for this kind comment.

Other places modified:

The text “Consisting of a dense layer, a convolutional layer, a recurrent neural network layer and other layers that use a nonlinear activation function and backpropagation,” in the third paragraph of the introduction in the original manuscript, has been removed to avoid duplication.

The text “USTC” in Fig. 3 has been adjusted in the revised manuscript for better illustration.

We changed the accuracy figure of “99.375%” to “99.38%” to keep to two decimal places.

The relevant section title was changed to “Comparison between deep learning method and the master equation” in the revised manuscript.

We changed “Figure” to “Fig(s).” where required in the revised manuscript.

The phrase “The deep learning device is a single NVIDIA GTX 1650 graphic processing unit with 4 GB of memory” in the Methods subsection entitled “Deep learning pipeline” has been removed to avoid duplication.

Figure 5 (e) and (f) are re-calculated based on the experiment parameters.

The codes for results including master equation fitting and deep learning prediction are given in a Github repository.

REVIEWER COMMENTS

Reviewer #1 (Remarks to the Author):

With this revision of their manuscript titled "Deep learning enhanced Rydberg multifrequency microwave recognition," the authors have markedly improved their manuscript and responded in a satisfactory way to all my previous comments. A detailed discussion of the effects of noise, including the spectrum and the types of noise investigated was added. The authors clarified my questions about the computational speed of the two methods of data analysis used and changed the manuscript accordingly. The methods section is now, in my opinion, more clear and more complete. The conclusions are more clear as well. Some figures have been changed to make them easier to understand, and some figures were added, mostly concerning the noise characteristics.

I think that the results presented here are very useful to other researchers in the field of AMO physics, because they can guide and inform future applications of deep learning to quantum-enhanced sensing with atomic systems. In my opinion, the modifications, added content, and clarifications make this new version of the manuscript much stronger.

I recommend publication in Nature Communications.

Reviewer #2 (Remarks to the Author):

The authors have in detail replied to the criticism of the first review round. The paper is much improved; in particular, the discussion of noise has become more quantitative, albeit still not at a level that the contributions of Master equation and machine learning can be compared on equal grounds.

However, my opinion on suitability has not changed, and I base my opinion on two main points.

1) The authors have given a detailed list of examples where machine learning has offered new insight into science. I appreciate this list, but I think the same does not apply to the present manuscript. A specific example that would provide deeper understanding through a neural network was, e.g., to unravel the noise contribution of external (microwave) noise from the intrinsic noise originating from atomic collisions—or, going even further, to extract the spectral function of collisional noise. This is not done, although I believe that the system is capable of doing so and it is, in fact, necessary to re-

formulate some claims. Instead, all physics information is cast away, leaving room for misinterpretation and overclaiming, see 2).

2) The Rydberg system employed is, in principle, very well understood. I have a hard time reading that a scientific publication on such a system celebrates that this deep physical understanding was not needed. By contrast, in my opinion, the claim that the neural network performs much better than the Master equation is based on a biased interpretation of the data.

Let me explain these statements. The fact that the Master equation apparently performs worse than a machine learning algorithm cannot be used, in my opinion, to support the initial claim for two reasons.

First, more sophisticated methods are available to include knowledge of the physical system. One example is a Bayesian analysis, which performs well even in systems with noise, probabilistically combining each new measurement data with the expected outcome. A balanced comparison would have used such a method. Since this has not been done, I believe the claims have to be toned down.

Second, in Fig. 4 the Master equation performs quite poor already for zero additional noise. Thus, collisional noise has a huge impact. This collisional noise is of course “learned” by the deep learning model. No surprise that the machine learning performs perfectly. Therefore, I believe that the comparison is not adequate and the statement on performance is overclaimed: the machine learning is given the knowledge of intrinsic noise, the Master equation is not. However, imagine using this very trained network now operating on a Rydberg tweezer array or an ultracold gas, without additional training, where intrinsic noise is (almost) absent. While the Master equation would perform almost perfectly for zero additional noise, the deep learning model would probably perform much worse, because the strong intrinsic noise is missing, and the network can usually not distinguish additional from missing noise.

Concluding, I still see the experimental work and the machine learning part as two decent pieces of work. I appreciate that signals can be transmitted and decoded even with admixed noise, when the structure of noise has been trained. But to make general and strong statements, I miss the final punch of either obtaining a novel insight into the system through machine learning, or proving a true advantage over state-of-the-art methods in an adequate comparison. The present claims are rather oversold in my view, and I still consider the use of neural networks here as just another application of machine learning as a black box.

Reviewer #3 (Remarks to the Author):

I am satisfied by the answers of the authors. I recommend publication in Nature Communications without further changes.

REVIEWER COMMENTS

Reviewer #1 (Remarks to the Author):

Reviewer Comment:

With this revision of their manuscript titled “Deep learning enhanced Rydberg multifrequency microwave recognition,” the authors have markedly improved their manuscript and responded in a satisfactory way to all my previous comments. A detailed discussion of the effects of noise, including the spectrum and the types of noise investigated was added. The authors clarified my questions about the computational speed of the two methods of data analysis used and changed the manuscript accordingly. The methods section is now, in my opinion, more clear and more complete. The conclusions are more clear as well. Some figures have been changed to make them easier to understand, and some figures were added, mostly concerning the noise characteristics.

I think that the results presented here are very useful to other researchers in the field of AMO physics, because they can guide and inform future applications of deep learning to quantum-enhanced sensing with atomic systems. In my opinion, the modifications, added content, and clarifications make this new version of the manuscript much stronger.

I recommend publication in Nature Communications.

Reply:

We thank the reviewer for reassessing our work in light of our previous exchange.

Reviewer #2 (Remarks to the Author):

Reviewer Comment:

The authors have in detail replied to the criticism of the first review round. The paper is much improved; in particular, the discussion of noise has become more quantitative, albeit still not at a level that the contributions of Master equation and machine learning can be compared on equal grounds. However, my opinion on suitability has not changed, and I base my opinion on two main points.

1) The authors have given a detailed list of examples where machine learning has offered new insight into science. I appreciate this list, but I think the same does not apply to the present manuscript. A specific example that would provide deeper understanding through a neural network was, e.g., to unravel the noise contribution of external (microwave) noise from the intrinsic noise originating from atomic collisions—or, going even further, to extract the spectral function of collisional noise. This is not done, although I believe that the system is capable of doing so and it is, in fact, necessary to re-formulate some claims. Instead, all physics information is cast away, leaving room for misinterpretation and overclaiming, see 2).

Reply:

We thank the referee for these valuable comments. We agree that more detail could be given. Whether the black box of neural networks can be opened or not remains an open question in the field of computer science. By following the suggestions from the referee, we try to open the black box and visualize the intermediate results of the deep learning model, from which we directly see how the deep learning model treats the transmission spectrum. The newly added results help us understand why the deep learning model performs so well and provide inspiration and new ideas for handling data obtained from the quantum sensor.

After passing through the max-pooling layer, the data treated in the deep learning model are visualized directly as shown in Figure C1. The data are expanded by the one-dimensional convolution kernel from one dimension to 20 dimensions. These new dimensions act as new features of the data. The feature space is then provided after the features of the intermediate results are reduced through principal component analysis (PCA) and T-distributed Stochastic Neighbor Embedding (t-SNE) [JMLR 12, pp. 2825 – 2830, 2011]. PCA is a method of linearly reducing the dimensionality to project data to a lower dimensional space whereas t-SNE is a method of nonlinearly reducing the dimensionality. The feature space is obtained after the intermediate results are fed into these new models (even a linear method such as PCA). Signals carrying the same message are inclined to be together whereas those carrying different messages are separated, as shown in Figure C2. The features captured by the deep learning model are thus transferable and compatible with other models. Figure C2 also presents dimensionality reduction results for noisy data, showing that the capturing of features by the deep learning model is robust against additional noise.

We refer to the Grad-CAM method, which is popular in the field of computer vision and used to highlight where the focus of a model is on an image [Int J Comput Vis 128, 336–359 (2020)]. Heat maps are presented in Figure C3, where the transmission spectrum and the model's focus are given. The area by which the deep learning model recognizes the signal from the heat maps is

obvious. This visualization method obtains a perspective of how the deep learning model handles the signal.

In short, the model has learned the feature space and to focus on special areas after being trained, which is one of the reasons for why the deep learning model performs so well.

Figure C1 Visualizations are conducted after data are output from the max-pooling (“MaxPooling1D”) layer.

Figure C2 Visualization of the intermediate results of multiple inputs. Dimensionality reduction methods are adopted. Each colored point represents a signal carrying a 3-bit message, and the same color data points carry the same message. This two-dimensional space is the feature space, where the points carrying the same message are clustered together and the points carrying different messages are separated. After 1-epoch (a) and 30-epoch (b) training, the data are fed into the model and the outputs of the max-pooling layer are handled in principal component analysis. In (c) another method (T-distributed Stochastic Neighbor Embedding) is adopted to deal with the intermediate results of the 30-epoch model. (d-f) t-SNE results with the standard deviation of additional noise $\sigma = 0.05, 0.75,$ and 0.95 .

Figure C3 Heat map of the model for singular data. The color represents how many focuses the model has on the local area of the data. The model recognizes the transmission spectrum via the area with bright color. (a) and (b) are heat maps for the data carrying messages “000” and “100”, respectively.

Manuscript changes:

- In page 4, we have added the description about how the deep learning model treats the transmission spectrum in the text: “This process is visualized to show how the deep learning model treats the transmission spectrum; more details are presented in the supplementary materials.”
- The Fig. 7 in main text has been revised. And “After the data output from the max-pooling layer, the visualizations are performed; see supplementary materials for more details.” has been added to the caption of Fig. 7.
- Those visualization results have been added to the supplementary materials.

Reviewer Comment:

2) The Rydberg system employed is, in principle, very well understood. I have a hard time reading that a scientific publication on such a system celebrates that this deep physical understanding was not needed. By contrast, in my opinion, the claim that the neural network performs much better than the Master equation is based on a biased interpretation of the data.

Let me explain these statements. The fact that the Master equation apparently performs worse than a machine learning algorithm cannot be used, in my opinion, to support the initial claim for two reasons.

First, more sophisticated methods are available to include knowledge of the physical system. One example is a Bayesian analysis, which performs well even in systems with noise, probabilistically combining each new measurement data with the expected outcome. A balanced comparison would have used such a method. Since this has not been done, I believe the claims have to be toned down.

Second, in Fig. 4 the Master equation performs quite poor already for zero additional noise. Thus, collisional noise has a huge impact. This collisional noise is of course “learned” by the deep learning model. No surprise that the machine learning performs perfectly. Therefore, I believe that the comparison is not adequate and the statement on performance is overclaimed: the machine

learning is given the knowledge of intrinsic noise, the Master equation is not. However, imagine using this very trained network now operating on a Rydberg tweezer array or an ultracold gas, without additional training, where intrinsic noise is (almost) absent. While the Master equation would perform almost perfectly for zero additional noise, the deep learning model would probably perform much worse, because the strong intrinsic noise is missing, and the network can usually not distinguish additional from missing noise.

Reply:

We thank the referee for this comment. We agree that, in principle, master equations with high-order terms and the whole noise spectrum can fit the experimental results perfectly. However, the equations then get too complex and nonlinear to solve when many microwave bins (more than 20) with tunable phases are involved. Researchers have often made approximations with master equations to obtain the approximated results; e.g., an approximation to the first order of Ω_p without considering the noise spectrum [PhysRevA.68.063801 (2003), PhysRevA.72.053801 (2005), PhysRevA.72.063814 (2005), PhysRevA.74.013812 (2006), PhysRevA.82.023812 (2010), PhysRevA.86.013815 (2012), PhysRevE.72.066703 (2005)].

Figure C4 shows the fitting of the theoretical and experimental curves with the master equation. If there is no internal or external noise, the master equation fits the spectrum perfectly, as shown in Figure C4 (a). However, when noise occurs in the experiment, the master equation performs poorly; see Figure C4 (b). Meanwhile, in Fig. 2(e) of the main text, we presented the performance of the deep learning model being trained and tested on the data with biased additional noise. In the last row of Fig. 2(e) of the main text, the model is trained on a training set without additional noise and is tested on a test set with additional noise that it has never seen before. However, the deep learning model performs well. Once the deep learning model has learned the signal pattern from the training set without additional noise, it extracts the features from the noisy test set even with the noise it has not seen before.

Indeed, to further increase the accuracy of the master equation, prior knowledge must be involved to limit the initial values. In contrast, the deep learning model learns to decode the signal by itself without human intervention. In the revised manuscript, to avoid confusion, we have toned down the claim about the master equation method and only say that the master equation that we employed is the commonly used one without considering the noise spectrum.

In our experiment, we separate the training and test processes just for simplicity. There are other deep learning methods, such as online learning, where the model adjusts its weights in real time after being trained and tested on the successively obtained data [arXiv:1711.03705 [cs.LG]]. The model could be more robust against noise after this training method. Additionally, it is unnecessary to retrain the whole model to generalize the deep learning model to other systems (e.g., cold atoms). All we need is to freeze the weights of most layers and tune the last few layers of the model on the new data set [ISBN: 978-1-61729-443-3].

Figure C4 Using the master equation to fit the theoretical curve (a) and experimental curve (b). The theoretical curve is generated by the master equation. In (a), the master equation fits the data well, whereas in (b), the master equation fits the data poorly.

Manuscript changes:

- We have toned down the claim by adding the sentence in the section “**Comparison between deep learning method and the master equation**”: “In our case, the master equation that we employed is the commonly used one without considering the noise spectrum.”
- The results about fitting theoretical (without noise) and experimental (with noise) transmission curve have been added in the supplementary materials.

Reviewer Comment:

Concluding, I still see the experimental work and the machine learning part as two decent pieces of work. I appreciate that signals can be transmitted and decoded even with admixed noise, when the structure of noise has been trained. But to make general and strong statements, I miss the final punch of either obtaining a novel insight into the system through machine learning, or proving a true advantage over state-of-the-art methods in an adequate comparison. The present claims are rather oversold in my view, and I still consider the use of neural networks here as just another application of machine learning as a black box.

Reply:

We thank the referee for the constructive comments. We have revised the manuscript following the suggestion of the referee to present a deeper understanding and the advantages of the deep learning model. The revisions are described as follows.

1. (a) We added the visualizations of the intermediate results, showing that the deep learning model learned to extract the features by learning from the data. The extracted features can be transferred to other models, which means that the deep learning model is a gray box (partially visualizable) and it has learned something understandable to other models. From the perspective of how the deep learning model handles the data, we have a new way of treating data.
 (b) We combined the deep learning model with Rydberg atoms to suppress the noise-induced effects due to the high sensitivity of the Rydberg atoms. (As noted by Reviewer #1, “the results presented here are very useful to other researchers in the field of AMO physics, because they can guide and inform future applications of deep learning to quantum-enhanced sensing with atomic systems.”)
2. Our work is the first to use the Rydberg atoms to receive and mix multi-frequency microwaves.

We provide a benchmark method by comparing with the commonly used master equation method. Additionally, we found that to further increase the accuracy of the master equation, prior knowledge must be involved to limit the initial values, whereas for deep learning, the model learns to decode the signal by itself without human intervention.

Reviewer #3 (Remarks to the Author):

Reviewer Comment:

I am satisfied by the answers of the authors. I recommend publication in Nature Communications without further changes.

Reply:

We thank the reviewer for reassessing our work in light of our previous exchange.

Other places modified:

In the abstract, “the information transmission rate is 900 kbps” has been changed to “the information transmission rate is 600 kbps”.

REVIEWERS' COMMENTS

Reviewer #1 (Remarks to the Author):

This new version of the manuscript is further improved compared to the previous version, which I already recommended for publication.

I applaud the authors for the addition of visualizations of intermediate results in the supplementary material. I think the visualizations are interesting and provide some insight into the inner workings of the deep learning algorithm.

I recommend publication in Nature Communications.

REVIEWER COMMENTS

Reviewer #1 (Remarks to the Author):

Reviewer Comment:

This new version of the manuscript is further improved compared to the previous version, which I already recommended for publication.

I applaud the authors for the addition of visualizations of intermediate results in the supplementary material. I think the visualizations are interesting and provide some insight into the inner workings of the deep learning algorithm.

I recommend publication in Nature Communications.

Reply:

We thank the reviewer for reassessing our work in light of our previous exchange.